# Upstream open reading frame translation enhances immunogenic peptide presentation in mitotically arrested cancer cells

Alexander Kowar [1,2,8], Jonas P. Becker [3,4,8], Rossella Del Pizzo[1,2], Zhiwei Tang [1,2], Julien Champagne [5], Kathrin Wellach [2,3,6], Kiana Samimi[2,3,6], Ariel Galindo-Albarrán [7], Pierre-René Körner [5], Jasmine Montenegro Navarro[5], Andrés Elía[1], Fiona Megan Tilghman[1], Hanan Sakeer [1], Marco Antonio Mendoza-Parra [7], Angelika B. Riemer [3,6] ✉, Reuven Agami [5] ✉ & Fabricio Loayza-Puch [1] ✉

Mitosis is a critical phase of the cell cycle and a vulnerable point where cancer cells can be disrupted, causing cell death and inhibiting tumor growth. Challenges such as drug resistance persist in clinical applications. During mitosis, mRNA translation is generally downregulated, while non-canonical translation of specific transcripts continues. Here, we show that mitotic cancer cells redistribute ribosomes toward the 5′ untranslated region (5′ UTR) and beginning of the coding sequence (CDS), enhancing translation of thousands of upstream open reading frames (uORFs) and upstream overlapping open reading frames (uoORFs). This mitotic induction of uORF/uoORF enriches human leukocyte antigen (HLA) presentation of non-canonical peptides on the surface of cancer cells after mitotic inhibitor treatment. Functional assays indicate these epitopes provoke cancer-cell killing by T cells. Our findings highlight the therapeutic potential of targeting uORF/uoORF-derived epitopes with mitotic inhibitors to enhance immune recognition and tumor cell elimination.

Mitosis is a critical phase of cellular reproduction and a key target for cancer therapy[1]. Although it is typically the shortest stage of the mammalian cell cycle, mitosis involves profound changes in cellular organization. The control of translation during mitosis plays an essential role in regulating the cell cycle, with particular significance in cancer biology. Most mRNAs undergo gene-specific translational downregulation during mitosis, rather than activation[2,3]. However, transcripts with a terminal oligopyrimidine (TOP) tract can escape this global translational suppression[4]. Cyclin-dependent kinase 1 (CDK1) plays an important role in regulating mRNA translation during the

---

[1]Translational Control and Metabolism, German Cancer Research Center (DKFZ), Heidelberg, Germany. [2]Faculty of Biosciences, University of Heidelberg, Heidelberg, Germany. [3]Division of Immunotherapy and Immunoprevention, German Cancer Research Center (DKFZ), Heidelberg, Germany. [4]Immuno-peptidomics Unit, National Center for Tumor Diseases (NCT), NCT Heidelberg, a partnership between DKFZ and University Hospital, Heidelberg, Germany. [5]Division of Oncogenomics, Oncode Institute, The Netherlands Cancer Institute, Amsterdam, the Netherlands. [6]Molecular Vaccine Design, German Center for Infection Research (DZIF), partner site Heidelberg, Heidelberg, Germany. [7]UMR 8030 Génomique Métabolique, Genoscope, Institut François Jacob, CEA, CNRS, University of Evry-val-d'Essonne, University Paris-Saclay, 91057 Évry, France. [8]These authors contributed equally: Alexander Kowar, Jonas P. Becker. ✉e-mail: a.riemer@dkfz.de; r.agami@nki.nl; f.loayza-puch@dkfz-heidelberg.de

M-phase of the cell cycle. Recent studies show that CDK1 influences translational regulation by phosphorylating specific substrates involved in mRNA translation. For example, CDK1 can phosphorylate ribosomal proteins and translation initiation factors, enhancing the protein synthesis needed for mitotic progression[5–7]. This activity ensures the cell possesses the necessary components for successful division.

Since cancer cells frequently bypass internal mitotic checkpoints, they often present high sensitivity to mitotic inhibitors[8]. These agents disrupt the microtubule dynamics required for chromosome segregation during cell division, leading to mitotic arrest and, ultimately, cell death[9,10]. Paclitaxel (Taxol), a widely used chemotherapeutic drug, acts by stabilizing microtubules and preventing their depolymerization[11,12]. It binds to the β-subunit of tubulin, locking the microtubule structure in place and inhibiting the dynamic reorganization of the microtubule network. This stabilization causes cell cycle arrest in the G2/M phase, which ultimately triggers apoptosis[13,14]. Clinically, paclitaxel is employed as both a first-line and subsequent therapy for several advanced carcinomas, including ovarian, breast, lung, and pancreatic cancers[15]. Its unique mechanism of action makes it effective against a broad range of malignancies, though resistance development remains a significant challenge in cancer treatment.

Non-canonical translation is increasingly recognized as a powerful means of diversifying the proteome and shaping the immunopeptidome[16,17]. Although early genome annotations largely ignored non-canonical open reading frames (ncORFs), due to concerns about false positives and insufficient validation methods[18], advances in ribosome profiling (Ribo-Seq) have transformed this view[19–24]. With near-codon resolution, Ribo-Seq now reveals actively translated regions across long noncoding RNAs, pseudogenes, and untranslated regions[25–30], uncovering thousands of ncORFs in human cells. Many of these hidden ORFs play important biological roles, regulating cell proliferation or generating neoantigens presented by major histocompatibility complex class I (MHC I; human leukocyte antigen (HLA) in humans)[31,32], and some encode microproteins essential for development and muscle function[33–35]. Yet despite their emerging importance, the potential for ncORF-derived peptides to serve as targets in combination with existing chemotherapies remains largely unexplored.

In this study, we investigate the dynamics of non-canonical translation in cancer cells arrested in mitosis. Using ribosome profiling, we observe a significant shift of ribosomes toward the 5′ UTR, leading to increased translation of uORFs and uoORFs. Additionally, we identify several uORF/uoORF-derived peptides presented on the surface of cancer cells, suggesting their potential role in shaping the immune response and serving as tumor antigens. Upon mitotic arrest induced by chemotherapy, we observe an enhanced presentation of specific uORF/uoORF-derived peptides in a murine model system, which are recognized by CD8$^+$ T cells, thus promoting tumor cell killing. Crucially, therapy-induced peptides identified in human cell lines elicit robust T cell responses in healthy donors. These findings underscore the potential of targeting therapy-induced uORF/uoORF-derived peptides as a promising approach in cancer immunotherapy, opening additional avenues for cancer treatment.

## Results

### Extensive ribosome redistribution in cancer cells arrested in mitosis

To investigate the translational dynamics of mitotically arrested cancer cells, we performed ribosome profiling on U-2 OS cells treated with Nocodazole, a drug that disrupts microtubule polymerization and halts cell cycle progression at mitosis (Fig. 1a). In proliferating cells, the distribution of ribosome-protected fragments (RPFs) was uniform across the coding sequence (CDS). By contrast, in mitotically arrested

cells, we observed a pronounced redistribution of ribosomes toward the 5′ UTR and the start of the CDS (Fig. 1b). Specifically, the proportion of RPFs mapping to the 5′ UTR increased approximately two-fold compared to the 3′ UTR (Fig. 1c), a pattern consistently seen across multiple cancer and non-tumorigenic cell lines (Supplementary Fig. 1a–i). Notably, although the 5′ UTR showed elevated RPF density, the 5′ CDS region exhibited an even higher density. This "front-loading" of ribosomes likely arises from enhanced initiation at the canonical start codon and possible early elongation pausing[36].

Nocodazole-induced microtubule depolymerization can cause protein misfolding and aggregation[37]. These aggregates can activate the integrated stress response (ISR), leading to eIF2α phosphorylation, which inhibits global protein synthesis while increasing translation from unconventional 5′ start sites[38]. To test whether ribosome redistribution is linked to stress-induced eIF2α phosphorylation, we treated U-2 OS cells with the eIF2α phosphatase inhibitor salubrinal and performed ribosome profiling. Although salubrinal treatment resulted in transcriptional upregulation of the stress-responsive genes *ASNS* and *CHOP* (Supplementary Fig. 1j), it did not induce a ribosome shift toward the 5′UTR (Supplementary Fig. 1k, l). Moreover, to determine whether the shift in footprints is drug- or mitotic arrest-specific, we induced mitotic arrest in U-2 OS cells through various molecular mechanisms. We treated U-2 OS cells with BI2536, a PLK1 inhibitor; *S*-trityl-ʟ-cysteine (STLC), an inhibitor of mitotic kinesin Eg5; or Taxol, a drug that stabilizes tubulin polymerization (Supplementary Fig. 1m). Translational activity was highly correlated across all mitotic arrest conditions (Fig. 1d). Ribosome profiling from all mitotic arrest treatments showed a similar extent of ribosome footprint redistribution and a concomitant increase in the proportion of RPFs in the 5′UTR (Fig. 1e, f), indicating that this redistribution is specific to mitotic arrest rather than a general stress response or drug-specific effect.

During mitosis, CDK1 substitutes for mTOR's role in activating cap-dependent translation by phosphorylating key translational regulators like the translation initiation factor eIF4E-binding protein 1 (4E-BP1)[2,5,6,39]. To test whether ribosome redistribution during mitotic arrest is dependent on mTOR activity, we treated mitotic U-2 OS cells with Torin 1, a potent and selective inhibitor of the mTOR kinase. Ribosome profiling showed that mTOR inhibition during mitosis has no effect on ribosome redistribution nor the relative proportion of RPFs in the 5′UTR (Fig. 1g, h). Furthermore, we observed that mTOR inhibition does not prevent the hyperphosphorylation of mTOR targets, such as 4E-BP1, which is known to play a key role in translation initiation (Supplementary Fig. 1n). These findings indicate that mTOR is not the primary driver of ribosome redistribution during mitosis and that other mechanisms, such as CDK1-mediated phosphorylation of 4E-BP1, may be more critical in this process.

Next, to determine whether the redistribution of RPFs in cells arrested in mitosis is associated with increased initiation rates at unconventional sites within the 5′ UTR, we performed global run-off ribosome profiling analysis using harringtonine. Harringtonine is a compound that specifically inhibits translation initiation by binding to the ribosomal machinery and preventing the formation of the first peptide bond. This results in the accumulation of ribosomes at translation start sites while depleting elongating ribosomes[40,41] (Fig. 1i). Given that the molecular phenotypes induced by different mitotic arrest agents (Fig. 1e, f) were consistent across all treatments, we used STLC to arrest cells in mitosis. In proliferating U-2 OS cells, most translation initiation sites (TISs) that accumulated over time were located at the annotated open reading frames (ORFs). In contrast, mitotically arrested U-2 OS cells showed an increase in initiation events at upstream translation initiation sites (uTISs), which levels were comparable to those at the annotated ORFs (Fig. 1j, k). A similar increase in 5′ UTR initiation events was observed in MDA-MB-231 cells arrested in mitosis (Supplementary Fig. 1o). Altogether, our data

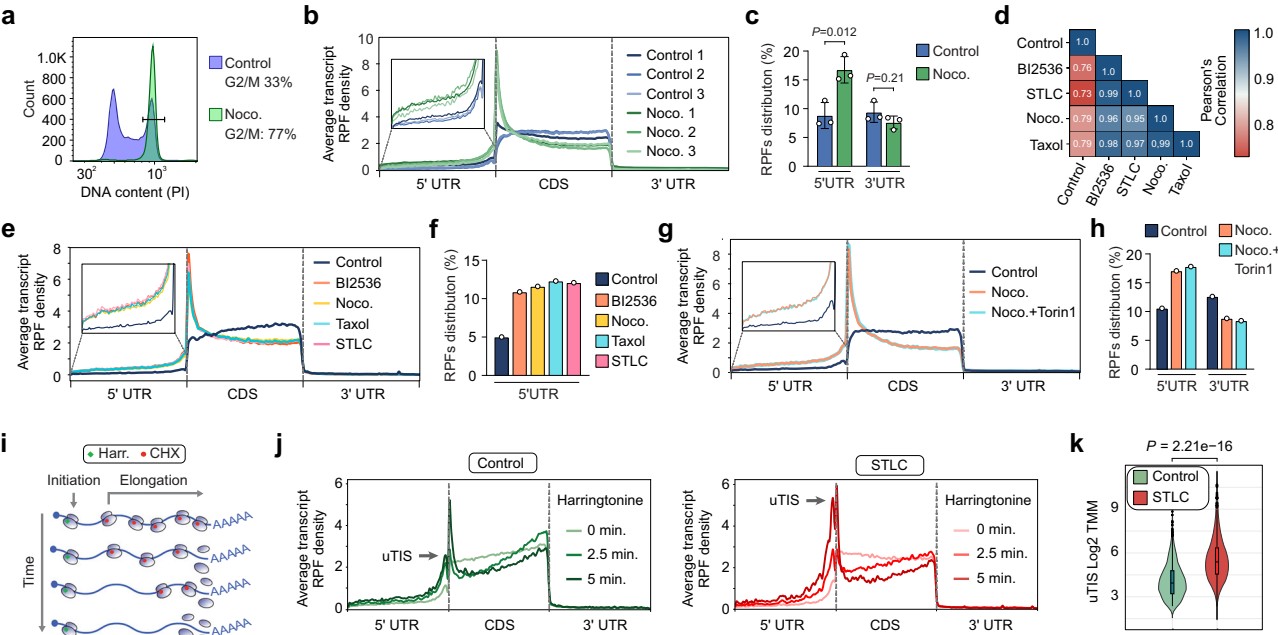

**Fig. 1 | Prolonged mitotic arrest leads to ribosome redistribution toward the 5' UTR. a** Representative propidium iodide (PI) staining of U-2 OS cells treated with nocodazole (0.5 μM, 16 h). The bar represents the percentage of cells in G2/M phase. Data from one experiment (n = 1). **b** Metagene profiles of ribosome-protected fragments (RPFs) in proliferating (blue) and mitotically arrested (green) U-2 OS cells (nocodazole, 0.5 μM, 16 h). Inset magnifies the 5' UTR. Data from biologically independent experiments (n = 3). **c** Quantification of RPFs distribution in the 5' UTR and 3' UTR of U-2 OS cells treated with vehicle or nocodazole (0.5 μM, 16 h). Mean ± SD from biologically independent experiments (n = 3); P values by two-tailed unpaired t-test. **d** Heatmap of Pearson correlation coefficients between gene RPKM in U-2 OS cells treated with vehicle (control), BI2536 (0.1 μM), noco-dazole (0.5 μM), taxol (1 μM), or STLC (5 μM) for 16 h. **e, f** Metagene profiles (**e**) and quantification (**f**) of RPFs in U-2 OS cells treated as in (**d**) for 16 h. Inset magnifies the

5' UTR. Data from one experiment (n = 1). **g, h**, Metagene profiles (**g**) and quanti-fication (**h**) of RPFs in U-2 OS cells treated with vehicle, nocodazole (0.5 μM, 16 h), or nocodazole (0.5 μM, 16 h) combined with torin1 (250 nM, 2 h). Inset magnifies the 5' UTR. Data from one experiment (n = 1). **i** Schematic of the run-off elongation experiment in U-2 OS cells. **j** Metagene profiles of RPFs in U-2 OS cells treated with vehicle or STLC (5 μM, 16 h) and harvested as described in (**i**). uTIS, upstream translation initiation site. Data from one experiment (n = 1). **k** Violin plots showing the trimmed mean of M values (TMM) distribution of uTIS in U-2 OS cells treated as in (**i**). Each point represents a TMM-normalized count at a predicted ORF site. Violin width indicates point density; center line is the median; upper and lower bounds are the 75th and 25th percentiles; whiskers represent minimum and maximum values. P value by one-sided Fisher test (no correction for multiple testing). Data from one experiment (n = 1). Source data are provided as a Source Data file.

suggest that prolonged mitotic arrest results in ribosome redistribu-tion, leading to increased unconventional initiation rates at the 5' UTR.

## Enhanced translation of uORF/uoORFs in mitotically arrested cancer cells

To systematically define the initiation events occurring in the 5'UTR of cells arrested in mitosis, we employed PRICE (probabilistic inference of codon activities by an EM Algorithm), a computational method spe-cifically designed for identifying non-canonical ORFs from ribosome profiling data[19] (Fig. 2a). After filtering out canonical coding sequences and truncated ORFs, we identified 1444 distinct actively translated non-canonical ORFs in proliferating cells and over 2600 in mitotically arrested U-2 OS cells treated with various agents (Supplementary Data 1–5). Notably, the proportion of actively translated uORFs and uoORFs more than doubled in mitotically arrested cells, regardless of the molecular mechanism inducing the arrest, while the proportion of other non-canonical ORFs remained stable (Fig. 2b). Similarly, nor-malizing the differential expression of RPFs on uORF/uoORFs revealed significant upregulation of most of the >1,000 uORFs/uoORFs during mitotic arrest (Supplementary Fig. 2a). This phenomenon was corro-borated by similar observations in PC3, MDA-MB-231, and RPE-1 cells treated with nocodazole (Supplementary Fig. 2b and Supplementary Data 6–11). Importantly, transient synchronization of U-2 OS in late G2, mitosis, or early G1 did not result in increased translation of uORF and uoORFs (Supplementary Fig. 2c, d). Similar results were observed in synchronized RPE-1 cells[2] (Supplementary Fig. 2e), indicating that prolonged mitotic arrest is required for the increased uORFs/uoORFs translation.

More than 80% of the over 1000 uORFs/uoORFs identified in mitotically arrested U-2 OS cells were commonly induced by at least two distinct molecular mechanisms (Fig. 2c). The uORFs/uoORFs exclusively translated in mitotically arrested cells were enriched in genes involved in biological processes such as cytoplasmic translation, mRNA splicing, and proteasome-mediated catabolic processes (Fig. 2d). These genes were also associated with cellular components like the cytoskeleton and focal adhesions (Supplementary Fig. 2f). Inter-estingly, the majority of predicted uORFs/uoORFs in proliferating U-2 OS cells initiated from non-ATG start codons (~84%). In contrast, a decrease to ~72% was observed in mitotically arrested cells, indicating a shift toward more canonical ATG initiation sites. The total number of uORFs/uoORFs with ATG initiation sites increased by at least 25% in mitotically arrested cancer cells (Fig. 2e and Supplementary Fig. 2g). We experimentally validated this observation using translation initia-tion sites sequencing (TIS-seq) in U-2 OS cells arrested in mitosis (Supplementary Fig. 2h).

To assess whether translation rates of these predicted uORF/ uoORFs were elevated in mitotically arrested cells, we calculated their translation efficiencies (TE) by normalizing RiboSeq reads mapped to genomic uORF/uoORFs coordinates against RNASeq reads from the same regions. Our analysis revealed that the vast majority of predicted uORF/uoORFs exhibited increased TE in mitotically arrested cells (Fig. 2f). Notable examples included PKM, MRPL51, and CAVIN1, which are implicated in critical cellular processes such as energy metabolism, mitochondrial function, and the oxidative stress response (Fig. 2g and Supplementary Fig. 2i). P-site-level visualizations demonstrate codon-level accumulation of ribosomes at predicted start sites, consistent

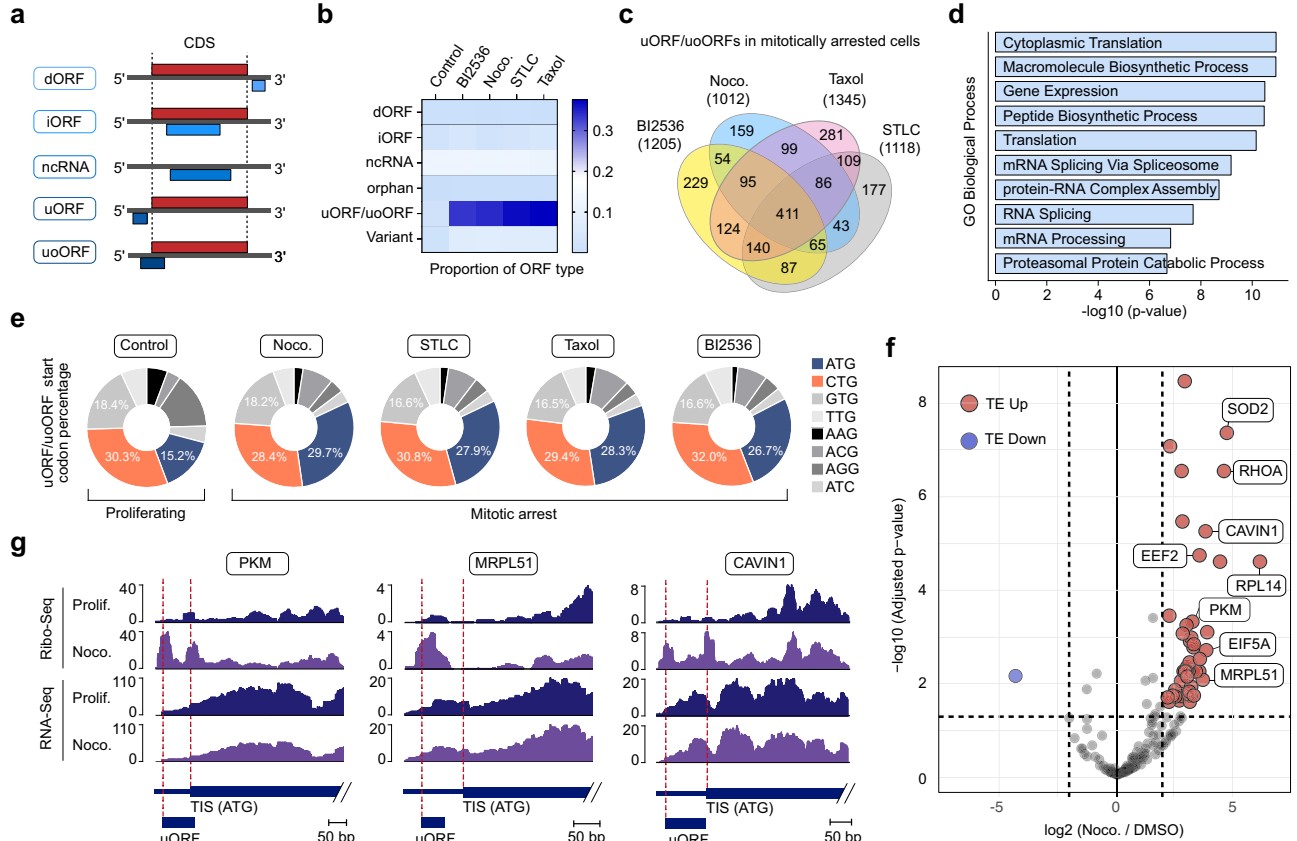

**Fig. 2 | Increased translation of uORFs/uoORFs in mitotically arrested cancer cells. a** Schematic overview of ORF types detected by PRICE. dORF downstream open reading frame, iORF internal open reading frame, ncRNA noncoding RNA, uORF upstream open reading frame, uoORF upstream overlapping open reading frame. **b** Proportion of ORFs of each type identified in U-2 OS cells treated with vehicle (Control), BI2536 (0.1 μM), Nocodazole (0.5 μM), Taxol (1 μM), or STLC (5 μM) for 16 h. "Variant" refers to ORFs that contain a stop codon but cannot be categorized as CDS or truncated CDS. "Orphan" refers to ORFs that do not fit into any of the described categories. **c** Venn diagram showing the number of uORFs/ uoORFs identified by PRICE in U-2 OS cells arrested in mitosis with BI2536 (0.1 μM), Nocodazole (0.5 μM), Taxol (1 μM), or STLC (5 μM) for 16 h. **d** Bar plot showing the combined scores for the GO database "Biological Process" obtained by Enrichr analysis of the common genes described in (**c**). *P* values were calculated using a one-

sided Fisher's exact test. Raw *P* values were adjusted using the Benjamini–Hochberg procedure. **e** Percentage of translation initiation site codons from uORFs/uoORFs in U-2 OS cells treated with vehicle (Control), BI2536 (0.1 μM), Nocodazole (0.5 μM), Taxol (1 μM), or STLC (5 μM) for 16 h. **f** Volcano plot illustrating the translation efficiency (TE) of all predicted uORFs/uoORFs in U-2 OS cells treated with 0.5 μM Nocodazole for 16 h. compared to asynchronous cells. The x-axis represents the log2 fold change in TE for uORFs/uoORFs. The y-axis indicates the significance of changes in TE. uORFs/uoORFs with increased TE are highlighted in red, while those with decreased TE are highlighted in blue. Data from two technical replicates. **g** Read distribution of representative uORFs/uoORFs with increased TE in mitotically arrested U-2 OS cells. RiboSeq (upper panels) and RNA-Seq (lower panels) reads are shown for the 5'UTR and the start of the CDS. Prolif proliferating, Noco Nocodazole.

with bona fide translation initiation (Supplementary Fig. 2j). Furthermore, mapping translation initiation sites (TIS) using Harringtonine in mitotically arrested cells demonstrated clear ribosome footprint peaks at predicted uORF/uoORFs start sites (Supplementary Fig. 2k). Collectively, our data indicate that ribosome redistribution during mitotic arrest significantly enhances the translation of thousands of uORF/ uoORFs.

### Immunopeptidomics identifies uORF/uoORF-derived peptides presented by HLA class I in cancer cells

Next, we investigated the presentation of uORF/uoORF-derived peptides on the surface of cancer cells via HLA class I molecules. To this end, we employed state-of-the-art liquid chromatography-tandem mass spectrometry (LC-MS/MS)-based immunopeptidomics on U-2 OS cells and the triple-negative breast cancer (TNBC) cell lines SUM-159PT and MDA-MB-231. For each cell line, three biological replicates of $5 \times 10^7$ cells were arrested in mitosis using Taxol. Taxol was chosen due to its ability to induce a stable mitotic arrest without causing the extensive cytotoxic effects or microtubule depolymerization associated with Nocodazole. Furthermore, Taxol's widespread use in

clinical settings makes it particularly relevant for translational applications. To support our analysis, we constructed a comprehensive mitotic uORF/uoORF database (uORF/uoORFdb) by selecting and translating nucleotide sequences of all uORFs and uoORFs identified by PRICE in the cell lines used in this study. This database, in combination with the annotated proteome, served as a reference for analyzing the immunopeptidomics dataset (Fig. 3a).

Our uORF/uoORFdb comprised 9,008 predicted ORFs (Supplementary Data 12), representing a significant expansion from previous studies. This extensive database allowed us to screen for more than 6700 uORFs and over 2200 uoORFs, substantially increasing the likelihood of identifying non-canonical ORF-derived HLA-presented peptides. Notably, our approach aligns with recent findings that uORF/ uoORFs can encode biologically active proteins and HLA-presented peptides in both malignant and benign cells, suggesting their potential role in cancer cell development and survival[42].

Our immunopeptidomics analysis revealed a substantial repertoire of HLA class I-presented peptides in U-2 OS, SUM-159PT, and MDA-MB-231 cells, with 12,904, 25,655, and 17,579 unique proteome-derived peptides identified, respectively. Notably, we discovered 127,

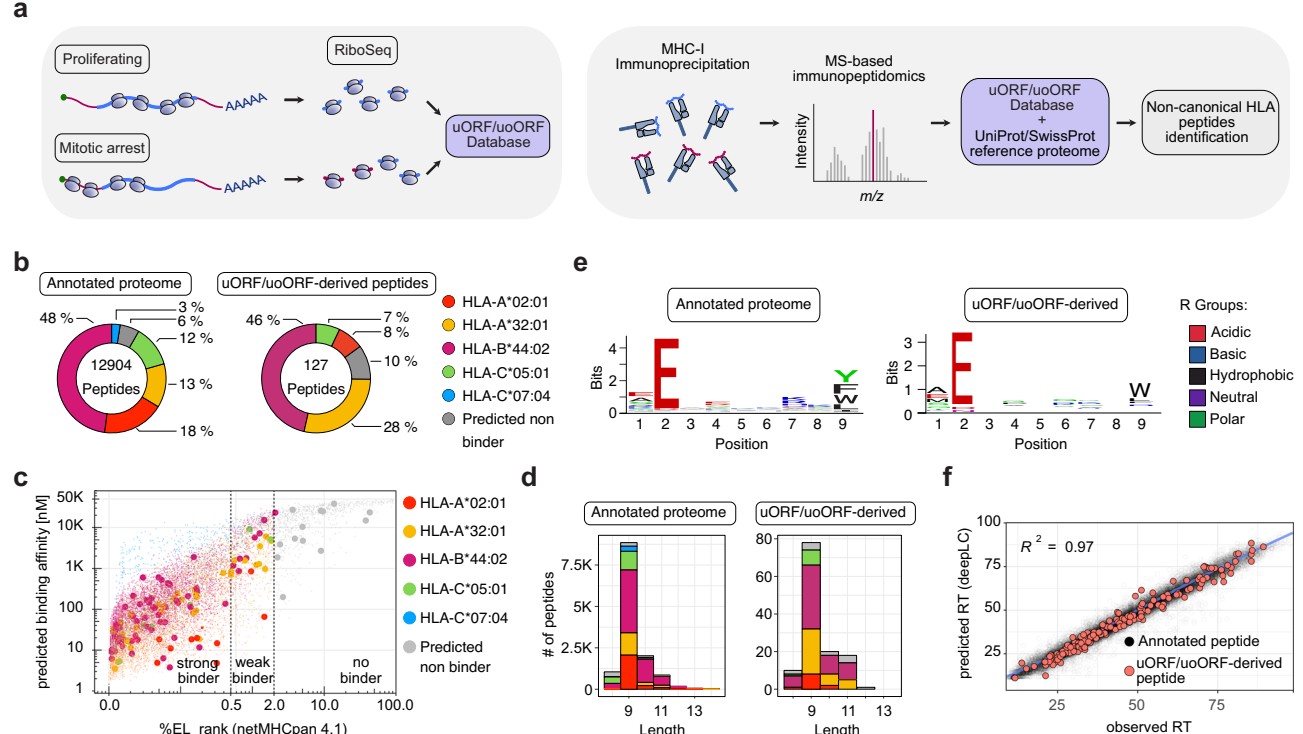

**Fig. 3 | uORF/uoORF-derived HLA-presented peptides exhibit comparable characteristics to annotated peptides. a** Schematic overview of the uORF/uoORF database creation using Riboseq and PRICE prediction, followed by the identification of non-canonical HLA peptides through liquid chromatography-tandem mass spectrometry (LC-MS/MS)-based immunopeptidomics. **b** Number of peptides mapping to the annotated proteome (left panel) and uORF/uoORF-derived proteome (right panel) of proliferating and mitotically arrested U-2 OS cells. The distribution of predicted binding to HLA alleles in U-2 OS cells is shown. Data from biologically independent experiments ($n = 3$) **c** Percentage of eluted ligand (EL) peptides predicted by NetMHCpan-4.1 plotted against predicted binding affinity for peptides derived from the annotated proteome (small dots) and uORF/uoORF-derived peptides (large dots) in proliferating and mitotically arrested U-2 OS cells. Predicted binding to HLA alleles in U-2 OS cells is shown. Peptides are categorized as strong binders (%EL rank 0–0.5), weak binders (%EL rank 0.5–2), or non-binders (%EL rank 2–100). Data from biologically independent experiments ($n = 3$) **d** Length distribution of detected peptides mapping to the annotated proteome (left panel) and uORF/uoORF-derived peptides (right panel) in proliferating and mitotically arrested U-2 OS cells. The proportion of predicted binding to U-2 OS HLA alleles is shown. Data from biologically independent experiments ($n = 3$) **e** Peptide motif plots of unique peptides from the annotated proteome (6624) and unique peptides derived from uORFs/uoORFs (58), confidently identified as binding to the U-2 OS allele HLA-B44:02. **f** Observed retention time (RT) plotted against predicted RT indices for peptides from the annotated proteome (black) and uORF/uoORF-derived proteome (red) across all HLA alleles in proliferating and mitotically arrested U-2 OS cells. $R^2$ represents the Pearson correlation coefficient. Data from biologically independent experiments ($n = 3$).

166, and 148 uORF-derived HLA class I-presented peptides in these cell lines (Fig. 3b and Supplementary Fig. 3a, f), representing 0.5–1% of the HLA-I immunopeptidome in mitotically arrested cells. Binding prediction revealed a similar distribution of assigned alleles for peptides derived from the canonical proteome and uORF/uoORF-derived peptides (Fig. 3b and Supplementary Fig. 3a, f). Predicted binding affinities to different HLA allotypes showed that 91% of uORF/uoORF-derived peptides and 90% of proteome-derived peptides were likely to bind to the HLA allotypes of the respective cell lines (Fig. 3c and Supplementary Fig. 3b, g). Both types of peptides, those from the canonical proteome and predicted uORF/uoORFs, exhibited the typical length distribution of HLA class I-presented peptides, predominantly as nonamers (Fig. 3d and Supplementary Fig. 3c, h). Sequence clustering of these peptides allowed reconstruction of the binding motifs for the HLA allotypes expressed by the respective cell line (Fig. 3e and Supplementary Fig. 3d, i). Moreover, predicted retention times for uORF/uoORF-derived peptides showed a high correlation with observed chromatographic retention times, comparable to those of annotated peptides (Fig. 3f and Supplementary Fig. 3e, j). Overall, the quality of uORF/uoORF-derived peptide identifications was comparable to that of annotated peptides, supporting their authenticity as genuine HLA class I-presented peptides.

To validate these identifications, we obtained stable isotope-labeled (SIL) versions of the top candidate peptides. We then performed parallel reaction monitoring (PRM) using optiPRM[43] on freshly prepared immunopeptidome extracts from U-2 OS, SUM-159PT, and MDA-MB-231 cells either without or with SIL spike-in, comparing the retention time and MS² fragmentation patterns of endogenous ("light") peptides with their SIL ("heavy") references. This approach confirmed co-elution and matching MS² spectra for nineteen unique uORF-derived peptides in U-2 OS cells, twenty-three in SUM-159PT cells, and eight in MDA-MB-231 cells (Supplementary Fig. 4), providing validation of our immunopeptidomic identifications.

Our results provide evidence that uORF/uoORF-derived peptides are presented by HLA class I molecules, suggesting that they may represent an underappreciated source of tumor antigens and potentially expand the repertoire of targets for cancer immunotherapy. This finding aligns with recent studies that underscore the significance of non-canonical peptides in the immunopeptidome[32,44,45].

## Identification of therapy-induced uORF/uoORF-derived peptides as potential antigenic targets in mitotically arrested cancer cells

Mitotic inhibitors offer a promising option for combination with checkpoint inhibitors or other immunotherapies to enhance immune response effectiveness[46]. However, the impact of mitotic arrest on the

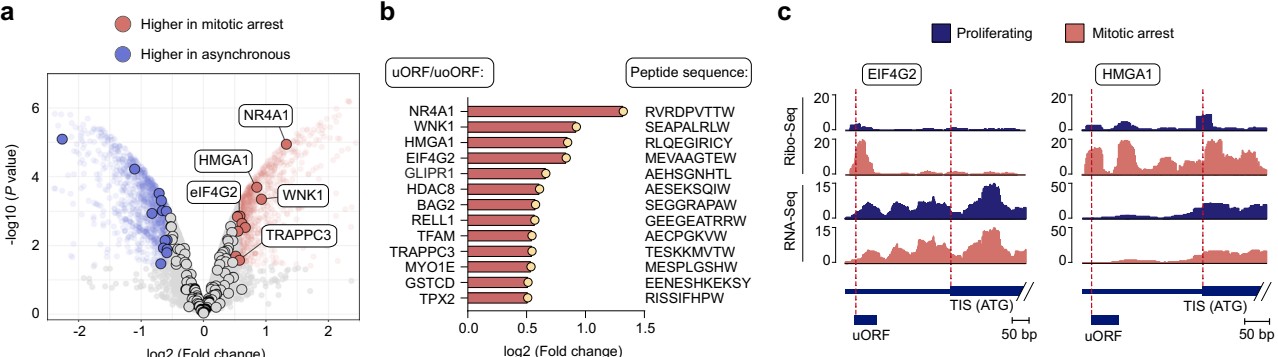

**Fig. 4 | Quantitative analysis of HLA-presented uORF/uoORF-derived peptides in mitotically arrested U-2 OS cells. a** Volcano plot illustrating the label-free quantification of the immunopeptidome in Taxol-treated versus DMSO-treated U-2 OS cells, with genes expressing uORF/uoORF-derived peptides highlighted. Peptides with a log2 fold change >0.5 and adjusted P value <0.05 are marked in red, while those with a log2 fold change <−0.5 and adjusted P value <0.05 are marked in blue. Statistical analysis was performed using an empirical Bayes moderated t-test with two-sided P values. Data from biologically independent experiments (n = 3).

**b** Name of the gene expressing the uORF/uoORF-derived peptide, log2 fold change (FC) of peptide abundance, and peptide sequence of mitotic arrest-induced peptides in U-2 OS cells. Data from biologically independent experiments (n = 3). **c** Read distribution of representative uORFs/uoORFs with increased expression in mitotically arrested U-2 OS cells. Ribo-Seq (upper panels) and RNA-Seq (lower panels) reads are displayed for the 5′ UTR and the start of the coding sequence (CDS).

HLA class I peptide repertoire remains uncharacterized. To address this, we aimed to quantify changes in uORF/uoORF-derived peptides in cancer cells following mitotic arrest in vitro, to better understand how mitotic inhibition could be leveraged in combination therapy regimens to improve patient outcomes.

To investigate alterations in the HLA-I-presented peptide repertoire induced by mitotic arrest, we performed label-free quantification of HLA-presented peptides, comparing Taxol-treated cells to DMSO-treated controls in U-2 OS, SUM-159PT, and MDA-MB-231 cell lines. We identified 13 peptides derived from uORFs/uoORFs with significantly higher HLA-mediated presentation in the immunopeptidome of mitotically arrested U-2 OS cells, 25 in mitotically arrested SUM-159PT cells, and 5 in mitotically arrested MDA-MB-231 cells (Fig. 4a, b and Supplementary Fig. 5a, b). Notably, these candidate peptides were associated with increased translation of their corresponding uORFs/uoORFs, despite no changes in RNA expression levels (Fig. 4c and Supplementary Fig. 5e). Furthermore, harringtonine-based mapping of translation initiation sites revealed an enrichment of ribosome-protected fragments at upstream initiation codons in mitotic cells (Supplementary Fig. 5c). While many of these sequences appear in ncORF databases (GENCODE ncORF, 36.6% overlap and nuORFdb, 26.1% overlap)[32,44], our immunopeptidomic data specifically reveal that their robust translation and HLA class I presentation are predominantly triggered under prolonged mitotic arrest rather than normal proliferation.

The genes expressing the upregulated uORF/uoORF immunopeptidome in mitotically arrested U-2 OS cells were associated with cellular components like the cytoskeleton and adherens junctions (Supplementary Fig. 5d). One of the detected peptides stems from a uORF in the *eIF4G2* gene, a non-canonical translation initiation factor that plays a crucial role in mitosis by facilitating the translation of specific mRNAs essential for cell division, including CDK1, which regulates phase transitions in the cell cycle[47]. Additionally, we identified a uORF-derived peptide from the *HMGA1* gene, a protein that significantly influences mitosis by modulating gene expression and chromatin structure. During cell division, HMGA1 promotes the transcription of genes involved in cell cycle progression, particularly those associated with the G2/M transition[48,49]. These uORF-derived peptides, characterized by their enrichment in mitotically arrested cells, represent promising antigenic targets for immunotherapy, aligning with recent findings that emphasize the role of non-canonical peptides in shaping immune responses[17,31].

## Enhanced presentation of uORF-derived peptides in mitotically arrested cancer cells facilitates targeted immune responses

Therapy-induced HLA-presentation of uORF/uoORF-derived peptides on the surface of cancer cells represents a promising opportunity for targeted immunotherapy. By displaying these peptides, mitotically arrested cancer cells following chemotherapy may acquire a unique immunogenic signature, making them potential candidates for precise immune targeting. To explore this possibility, we generated luciferase reporters containing 5′ UTRs with uORFs that exhibited increased peptide presentation in mitotically arrested cells. We replaced the immunopeptidomics-identified sequences with the SIINFEKL peptide from chicken ovalbumin (OVA), an eight-amino-acid peptide presented by H2-K$^b$. This complex is recognized by T cell receptor (TCR)-transgenic CD8$^+$ OT-I T cells, as well as by a TCR-like antibody (clone 25-D1.16), facilitating the assessment of SIINFEKL epitope presentation via flow cytometry[50]. For this analysis, we specifically selected the 5′ UTRs of eIF4G2 and TPX2, each containing uORFs with start and stop codons located entirely upstream of the annotated CDS, making them suitable for our reporter system (Fig. 5a). As a negative control, we used the 5′ UTR of GAPDH, which has no predicted uORFs.

Next, we assessed whether therapy-induced mitotic arrest enhances the presentation of uORF-derived peptides in cancer cells. We transfected the murine cancer cell line TC-1 with uORF-SIINFEKL reporters and the GAPDH negative control. In proliferating cells, reporter expression did not lead to significant recognition by the 25-D1.16 antibody (Fig. 5b and Supplementary Fig. 6a). In contrast, mitotically arrested cells exhibited a marked increase in uORF-derived SIINFEKL presentation, regardless of the agent used to induce arrest (Fig. 5b and Supplementary Fig. 6b). Although luciferase activity decreased during mitotic arrest, mRNA levels remained unchanged (Supplementary Fig. 6c, d). To further investigate the role of uORF translation during mitotic arrest, we mutated the uORF start codons from ATG to ATA. TC-1 cells expressing these mutant reporters showed reduced SIINFEKL presentation upon mitotic arrest, indicating that active uORF translation during mitosis is crucial for effective antigen presentation (Fig. 5c).

To test whether the increased presentation of uORF-derived peptides during mitotic arrest enhances CD8$^+$ T cell-mediated cytotoxicity, we activated OT-I CD8$^+$ T cells with anti-CD3, anti-CD28, and IL-2 for 72 h and co-cultured them with either proliferating or mitotically arrested TC-1 cells expressing the uORF-SIINFEKL reporters and the GAPDH negative control (Fig. 5d). Neither transfection nor mitotic

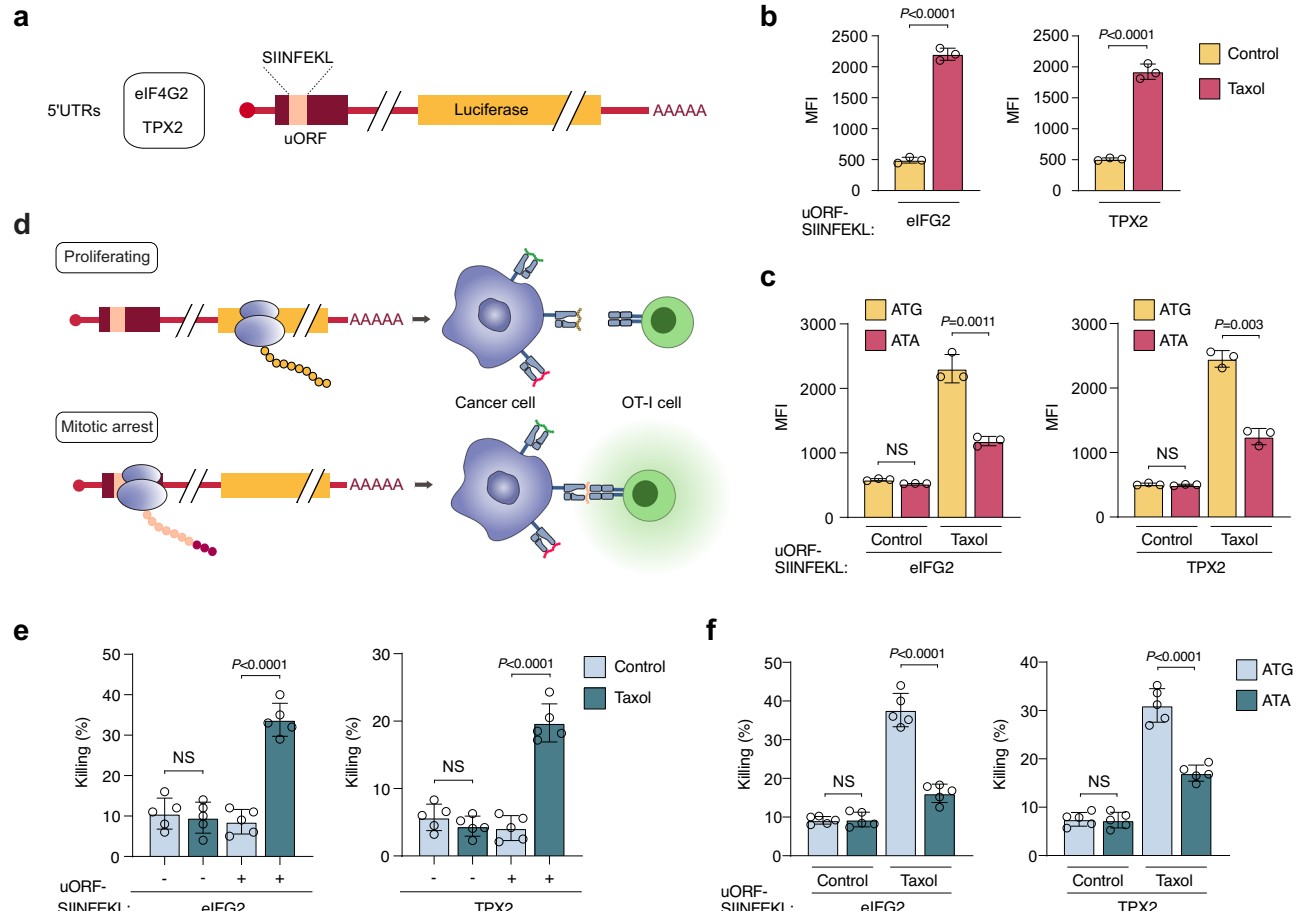

**Fig. 5 | Increased uORF-derived peptide presentation in mitotic cancer cells promotes targeted immune responses. a** Schematic of the uORF reporter system. Two different 5′ UTRs (eIF4G2 and TPX2) were selected and cloned into the pGL3-promoter vector. The sequence of the uORF-derived peptide was replaced with the SIINFEKL sequence, followed by firefly luciferase. **b** Detection of the p:MHC complex SIINFEKL:H-2Kᵇ by flow cytometry, shown as Median fluorescence intensity (MFI), in TC1 cells transfected with the indicated reporters. Cells were treated with Taxol (1 μM) or DMSO (Control) for 16 h. Data represent mean ± SD from biologically independent experiments ($n = 3$). *P* values were calculated using a two-tailed unpaired *t*-test. **c** MFI of the SIINFEKL peptide bound to H-2Kᵇ in TC1 cells transfected with wild-type uORF-SIINFEKL reporters or start codon mutant uORF-SIINFEKL reporters. Cells were treated with Taxol (1 μM) for 16 h. Data represent

mean ± SD from biologically independent experiments ($n = 3$). *P* values were calculated using a two-tailed unpaired *t*-test. **d** Schematic illustrating how the presentation of uORF-derived SIINFEKL peptides induced by mitotic arrest leads to OT-I T cell recognition. Proliferating cells expressing uORF-SIINFEKL reporters are not recognized by OT-I cells, whereas enhanced uORF translation in mitotically arrested cells allows for their recognition. **e, f** Percentage of TC1 cell killing in vitro by activated OT-I cells. TC1 cells were transfected with the indicated reporters and arrested in mitosis with Taxol (1 μM) for 16 hrs. or treated with DMSO for the same period. Data represent mean ± SD from biologically independent experiments ($n = 5$). *P* values were calculated using a two-tailed unpaired *t*-test. NS non-significant. Source data are provided as a Source Data file.

arrest alone resulted in significant cytotoxic effects (Fig. 5e and Supplementary Fig. 6e). However, mitotic arrest in reporter-expressing cells significantly enhanced cytotoxic activity and tumor cell killing by OT-I T cells in vitro (Fig. 5e). Consistent with these observations, OT-I T cells secreted IFN-γ only after co-culture with mitotically arrested cancer cells expressing the uORF-SIINFEKL reporters (Supplementary Fig. 6f). Notably, cancer cells expressing reporters with mutant start codons showed reduced antigen-specific killing by OT-I T cells, accompanied by a significant decrease in IFN-γ secretion during co-culture (Fig. 5f and Supplementary Fig. 6g). These findings highlight the potential of therapy-induced uORF-derived peptides as innovative targets for immunotherapy, underscoring their role in enhancing immune recognition during cancer treatment.

To assess the immunogenicity of treatment-induced, non-canonical peptides identified in the human cancer cell lines, we successfully conducted in vitro priming and expansion of peptide-specific cytotoxic T lymphocytes for two uORF/uoORF-derived peptides, REMFIWAVA and CSKVSSEY. Briefly, peripheral blood mononuclear cells (PBMCs) from healthy donors expressing the matching

HLA allotypes were stimulated in vitro with the respective peptide and cytokine cocktails for APC maturation and activation. Following expansion, we observed robust peptide-specific IFN-γ secretion upon stimulation with REMFIWAVA and CSKVSSEY in ELISpot assays, whereas the same-length control peptides did not trigger a response (Fig. 6a, b, d, e and Supplementary Fig. 7c–f). Epitope-specific activation of polyfunctional CD8⁺ T cells was additionally validated by their production of IFN-γ and TNF-α upon peptide challenge (Fig. 6c, f), while no CD4⁺ T cell reactivity could be detected (Supplementary Fig. 7a, b). REMFIWAVA even showed reactivity in two independent healthy donors. These data confirm that the uORF/uoORF-derived peptides, REMFIWAVA and CSKVSSEY, are capable of eliciting antigen-specific CD8⁺ T cell responses, highlighting their potential as targets for immunotherapy.

## Discussion

Our study reveals that mitotic arrest induced by agents such as Paclitaxel or Nocodazole dramatically reshapes the translational landscape of cancer cells, redirecting ribosomes toward the 5′UTR and early

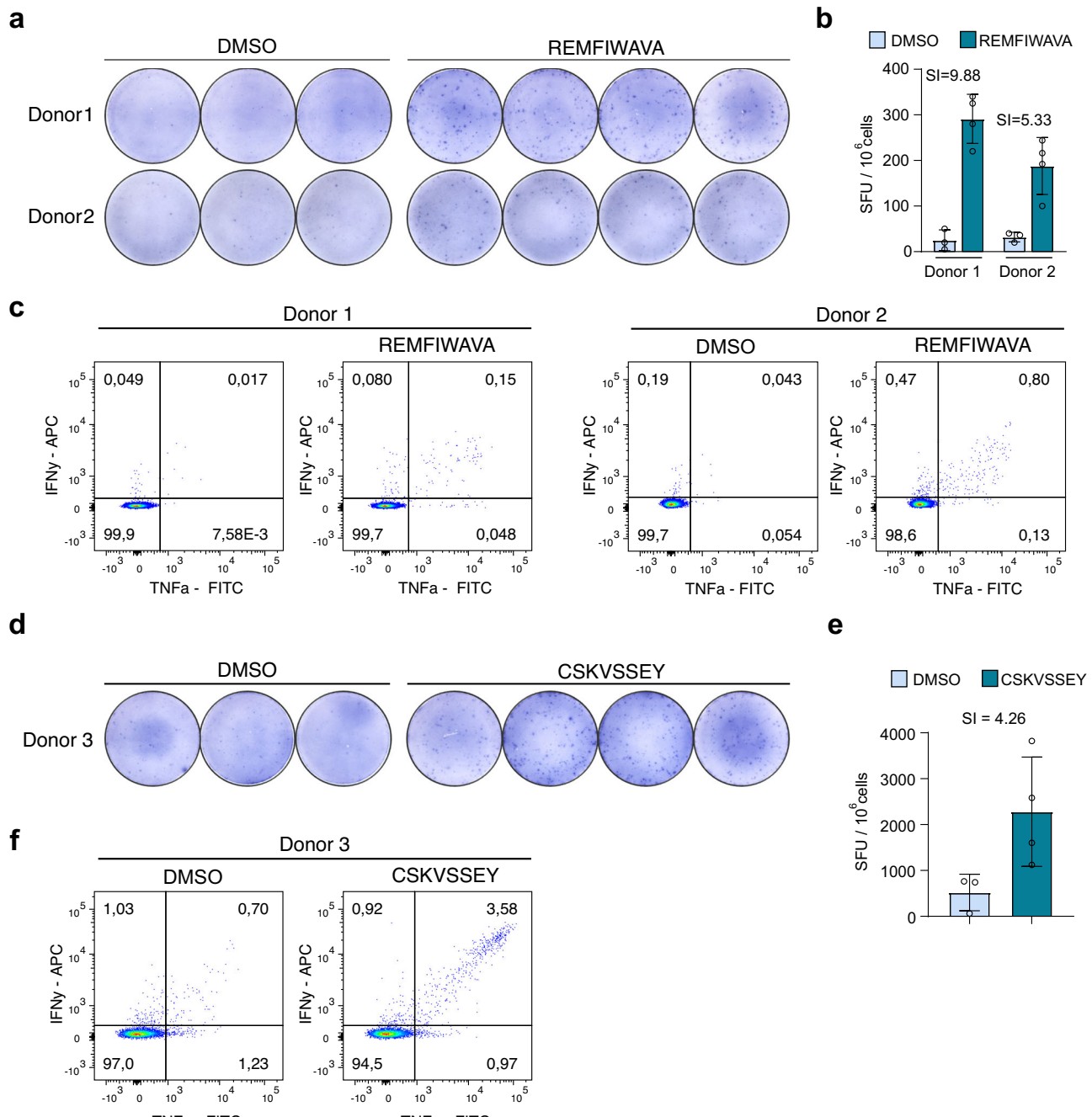

**Fig. 6 | Robust T cell immunity against uORF-derived peptides in healthy donors. a** Representative ELISPOT of PBMCs from two healthy donors stimulated with the uORF-derived peptide REMFIWAVA, with DMSO as the negative control. **b**, ELISPOT quantification from panel (**a**). Data were presented as spot-forming units (SFU) per $10^6$ PBMCs and represent mean ± SD from technical replicates ($n = 3$ for DMSO treatment or $n = 4$ for peptide treatment). SI stimulation index. **c** IFN-γ and TNF-α expression in CD8$^+$ cells from two healthy donors treated with DMSO or the uORF-derived peptide REMFIWAVA. **d** Representative ELISPOT showing PBMCs from a single healthy donor stimulated with the uORF-derived peptide CSKVSSEY and compared against a DMSO negative control. **e** Quantification of the ELISPOT results displayed in (**d**). Data were expressed as spot-forming units (SFU) per $10^6$ PBMCs, presented as the mean ± SD from technical replicates ($n = 3$ for DMSO; $n = 4$ for the peptide). SI refers to the stimulation index. **f** Expression of IFN-γ and TNF-α in CD8$^+$ T cells from the same single donor after treatment with DMSO or the uORF-derived peptide CSKVSSEY.

coding regions. This redistribution is driven by CDK1-mediated phosphorylation of 4E-BP1[5,6], resulting in substantially increased initiation at upstream open reading frames (uORFs) and overlapping uORFs (uoORFs). We speculate that this shift also reflects the integration of cell-cycle checkpoint signaling, stress-response pathways, and the selective modulation of key initiation factors, which together favor uORF usage during prolonged M-phase arrest. By integrating ribosome profiling, immunopeptidomics, and functional assays, we demonstrate

that thousands of these translated, non-canonical peptides are displayed on the surface of mitotically arrested cells via HLA class I molecules, where they can be recognized by CD8$^+$ T cells.

These findings align with recent work by Ly et al., which highlights how mitotic translational shifts can profoundly expand the proteome[51]. Our data further underscore the therapeutic potential of uORF/uoORF-derived peptides: we show that they elicit immune recognition and drive T cell-mediated lysis of drug-treated cancer cells,

indicating their value as anti-tumor targets. Notably, none of the peptides we identified were found in existing immunopeptidomics or proteomics databases, underscoring their treatment specificity; we term them "therapy-induced uORF/uoORF-derived epitopes." To capture as many relevant sequences as possible, we consolidated PRICE-predicted ORFs from multiple cell lines, thereby creating a combined database that likely contains the majority of these cryptic peptides. Although cell-type–specific uORFs may remain unrepresented, our approach provides a robust starting point for discovering non-canonical, therapy-induced antigens.

Therapy-induced uORF/uoORF-derived peptides offer an exciting avenue for cancer immunotherapy, especially when coupled with mitotic inhibitors like Paclitaxel. When assessing the immunogenicity of these epitopes in PBMCs from healthy human donors, we observed robust immune reactivity in two independent donors. Consequently, these therapy-induced epitopes could serve as tumor antigens. Beyond its role in stabilizing microtubules, Paclitaxel also promotes immunogenic cell death (ICD), enabling dendritic cells and macrophages to amplify anti-tumor responses[52,53]. In parallel, it augments Th1 immunity by boosting IFN-γ–secreting CD8$^+$ T cells and IL-2–producing CD4$^+$ T cells[54]. and restrains immunosuppressive populations such as MDSCs and Tregs[11]. Notably, clinical analyses of breast cancer specimens indicate that overall immune cell infiltration remains largely unchanged following Paclitaxel treatment, suggesting its immunomodulatory impact stems from enhanced antigen presentation rather than altered immune abundance (Supplementary Fig. 9f). Consequently, combining Paclitaxel with immune-based interventions could leverage both the tumoricidal effects of mitotic arrest and the non-canonical immunopeptidome to enhance tumor clearance.

An important consideration is the variability in therapy-induced non-canonical peptides across different tumor subtypes and disease stages[55]. Genetic and epigenetic alterations, as well as distinct translational landscapes, could influence the breadth and immunogenicity of these peptides. While some are broadly shared, enabling cross-tumor immunotherapy strategies, others might be unique to specific cancer types, thereby supporting a precision immunotherapy approach[56]. Future investigations should systematically map the therapy-induced immunopeptidome across diverse cancers, correlating peptide profiles with clinical responses and potential immune escape mechanisms[57,58].

Finally, although our cell line–based system is well-suited to dissect translational events and confirm T cell recognition in vitro, it cannot fully capture the complexity of the tumor microenvironment or account for interpatient heterogeneity. Clinical or patient-derived xenograft samples will be essential for determining how well these findings translate to patients undergoing mitotic inhibitor therapy. Moreover, deeper exploration of near-cognate initiation sites, synergy with additional immunotherapies (e.g., checkpoint blockade), and in vivo validation of immunogenicity will further define the clinical potential of these therapy-induced peptides.

Overall, our work provides a blueprint for leveraging therapy-induced translational remodeling to expand the repertoire of targetable antigens in cancer. By harnessing non-canonical uORF/uoORF-derived peptides unveiled during prolonged mitotic arrest, we open additional possibilities for combination treatments that unite the cytotoxic power of mitotic inhibitors with the specificity of T cell-mediated immunity.

## Methods
### Cell culture
U2OS (ATCC, HTB-96), SUM-159PT (BioIVT, CVCL_5423), MDA-MD-231 (ATCC, HTB-26), PC3 (ATCC, CRL-1435), TC1 (ATCC, CRL-2785), and RPE-1 (ATCC, CRL4000) cell lines were cultured in DMEM High Glucose (Thermo Fisher Scientific) supplemented with 10% FBS

(Thermo Fisher Scientific) and 1% PenStrep (Thermo Fisher Scientific) at 37 °C and 5% CO$_2$. For mitotic shake-off experiments, cells were cultured to a confluency of 70% and treated with mitosis-arresting compounds for 16 h; BI2536, 0.1 μM (Cell Signaling Technology, #26744), Nocodazole, 0.5 μM (Sigma-Aldrich, M1404), Taxol, 1 μM (Santa Cruz, sc-201439), STLC 5 μM (Tocris, #2191). The medium was centrifuged at 600 × g for 10 min at 4 °C. The cell pellets were washed in ice-cold PBS. All cell lines were regularly tested for *Mycoplasma* contamination. All small molecules used in this study comply with the Chemical Probes Portal criteria (https://www.chemicalprobes.org/information-centre): each compound has been validated for target potency (IC$_{50}$/K_d values within the recommended range), demonstrated high selectivity over off-target proteins, and is accompanied by appropriate negative controls.

### Propidium iodide (PI) staining
Cells were resuspended in PBS containing 1 mM EDTA, and counted using a Casy Counter system. The cell concentration was adjusted to ensure equal cell numbers. The cells were then fixed in ice-cold absolute ethanol and stored overnight at −20 °C. Following storage, the cells were washed three times with PBS + 1 mM EDTA and resuspended in PI staining buffer (50 μg/ml PI, 0.2 mg/ml RNase A, 0.4% Triton X-100, 1 mM EDTA in PBS). The cells were incubated at 37 °C for 30 min. with constant shaking. After incubation, the samples were filtered into round-bottom tubes containing cell strainers, and analyzed using a BD Canto II system. To select PI-stained single cells, the following gating strategy was applied: cells were initially separated using FSC and SSC to distinguish the cell population. Single cells were then identified by combining FSC-H and FSC-A, followed by 488 nm excitation and filtering for the BL84/42 signal.

### Cell synchronization
U-2 OS cells were synchronized in the G2 phase by treatment with the CDK1 inhibitor RO-3306 (6 μM) for 18 h. After synchronization, cells followed one of two experimental routes: they were either harvested immediately to obtain the G2 sample, or the inhibitor was removed to allow mitotic entry. Mitotic cells were isolated by mechanical shake-off 45 min. after RO-3306 washout and either collected immediately for the mitotic sample or re-plated to allow cell cycle progression into G1. These post-mitotic cells were then harvested 3 h after re-plating to generate the G1 sample.

### Immunoblotting
Cell pellets were harvested and lysed in whole-cell lysis buffer (1 M Tris, pH 7.5, 10% glycerol, 2% SDS). Next, 1 μl of 0.1 M MgCl$_2$ solution and 2.5 U of Benzonase (Merck) was added and incubated at 37 °C for 10 min. to digest DNA. Bradford assay (Serva, Bradford reagent 5x) was used to determine the level of protein input. Lysates were diluted with water and 4x Laemmli Buffer (Bio-Rad) supplemented with 10% β-mercaptoethanol. Samples were separated by SDS-PAGE and transferred to nitrocellulose membranes using TurboBlot (Bio-Rad) with transfer buffer (240 mM Tris-HCl, 195 mM Glycin, 0.5% SDS). Next, membranes were blocked in TBS-T plus either 5% non-fat dry milk or 5% BSA for 1 h at room temperature. Primary antibodies were diluted in TBS-T plus either 5% non-fat dry milk or 5% BSA and incubated with the membrane overnight at 4 °C and gentle rotation. Next, membranes were washed four times with TBS-T for 5 min at room temperature before incubating with the secondary antibody in TBS-T for 1 h at room temperature, protected from light. Membranes were imaged using an Odyssey CLx machine (LICORbio). Antibodies used were 4E-BP1 (Cell Signaling; cat. #9644, 1:1000), Phospho-4E-BP1 (Thr37/46) (Cell Signaling; cat. #2855, 1:1000), Phospho-Histone H3 (Ser10) (Cell Signaling; cat. #9701, 1:1000), GAPDH (ProteinTech; cat. #60004; 1:20,000).

## RiboSeq

Cells were resuspended in lysis buffer (20 mM Tris-HCl, pH 7.5, 10 mM MgCl$_2$, 100 mM KCl, 1% Triton X-100), supplemented with 100 μg/mL cycloheximide (CHX), 2 mM DTT, and 1x Complete Protease Inhibitor (Roche). They were then treated with RNase I (Ambion) at a concentration of 1.2 U/μL for 45 min at room temperature. The samples were layered onto sucrose gradients, ranging from 47 to 7% sucrose, prepared in 20 mM Tris-HCl, pH 7.5, 10 mM MgCl$_2$, 100 mM KCl, 100 μg/mL CHX, and 2 mM DTT. The gradients were centrifuged for 2 h at 238,000 × $g$ (36,000 rpm) at 4 °C using a Beckman-Coulter ultracentrifuge with an SW41-Ti rotor. Monosome-containing fractions were collected and digested with 15 μL of recombinant Proteinase K (Roche) and 1% SDS for 45 min at 45 °C. Subsequently, RNA was extracted using a standard phenol-chloroform-guanidinium thiocyanate method. The resulting ribosome-protected fragments (RPFs) were size-selected (20-34 nt) and 3′ dephosphorylated using T4 Polynucleotide Kinase (PNK; New England Biolabs). Following this, 5′ pre-adenylated linkers were ligated to the 3′ ends of the RPFs using T4 RNA Ligase 2 (New England Biolabs), and rRNA depletion was performed using custom biotinylated rRNA-oligonucleotides and streptavidin-coated magnetic beads. Reverse transcription was carried out using the SuperScript III First-Strand Synthesis Kit (Thermo Fisher). cDNA was circularized using CircLigase ssDNA Ligase II (LGC Biosearch Technologies) and then amplified via PCR using Q5 High-Fidelity 2x Master Mix (New England Biolabs). DNA was quantified using the Qubit dsDNA HS Kit and adjusted to a concentration of 2 nM, suitable for NextSeq 2000 sequencing. All oligo sequences used in this study and quality control analysis are shown in Supplementary Data 13 and Supplementary Fig. 9, respectively.

## Harringtonine assay

Cells were treated with 2 μg/mL harringtonine at the indicated time points before harvest. For all experiments, cycloheximide (CHX, 100 μg/mL) and matching DMSO concentrations were included in both control and treatment conditions, with harringtonine added only in the treatment condition. Control cells were scraped into the medium, while mitotically arrested cells were harvested by shake-off and collected from the medium. All samples were then pelleted and resuspended in lysis buffer (20 mM Tris-HCl, pH 7.5, 10 mM MgCl$_2$, 100 mM KCl, 1% Triton X-100), supplemented with CHX (100 μg/mL), 2 mM DTT, and 1× Complete Protease Inhibitor (Roche). This design ensures that CHX-only samples serve as the appropriate control, and all samples experience the same duration of harringtonine treatment.

## RiboSeq analysis

Sample adapters were trimmed using cutadapt (v3.4) and demultiplexed with barcode_splitter from FASTX-toolkit (v0.0.6). Fragments smaller than 20 nt were dropped. UMIs extraction was performed using umi_tools (v1.1.1). By the BLAST-like alignment tool (BLAT) (v36x2), rRNA reads were filtered and discarded. The rRNA index for RNA18S5, RNA28S5 and RNA5-8S5 was constructed manually from NCBI RefSeq annotation. Remaining reads were aligned with Spliced Transcripts Alignment to a Reference (STAR) (v2.5.3a) to GRCh37/hg19 with --outSAMtype BAM Unsorted --readFilesCommand zcat --quantMode TranscriptomeSAM GeneCounts --outSAMmapqUnique 0. Genome browser bigwig tracks were obtained using samtools (v1.15.1) and bedtools (v2.24.0).

## Transcript distribution

To analyze ribosome transcript distribution, we selected the most representative isoform for each gene using a hierarchical selection strategy. RPF counts were obtained for each selected transcript, and intra-gene normalization was performed by dividing the cumulative read counts for each region (5′ UTR, CDS, 3′ UTR) by the total RPF counts for that transcript. This normalization allowed for comparisons across different regions of the same transcript. Next, we computed RPF density across regions for each transcript by interpolating read counts over a fixed grid of 2000 points. Transcripts with fewer than 50 reads were excluded from the downstream analysis. The interpolated RPF densities across transcripts were then averaged and subjected to Gaussian smoothing to reduce noise. Finally, the resulting RPF densities were plotted alongside the corresponding transcript regions.

## Upstream translation initiation sites quantification

Predicted uORF and uoORF genomic coordinates from U-2 OS RiboSeq data were compiled in SAF format and counted using featureCounts from the subread package (v1.5.1). Resulting counting tables were filtered for ≥5 reads per sample and feature. Next, the trimmed mean of $M$ values (TMM) for cross-sample comparison were determined using the "calcNormFactors" function from edgeR. Normalized counts per million (cpm) were subjected to sample-specific outlier calculation using the Grupps function.

## ORF prediction

ORFs were predicted using PRICE[19]. In brief, UMI-extracted and rRNA-filtered FASTQ files were re-aligned to the GRCh37/hg19 reference genome using STAR (v2.5.3a). Important outSAMattributes required by PRICE were specified, including --outSAMtype BAM Unsorted, --alignEndsType Extend5pOfReads12, --outSAMattributes nM MD NH, --readFilesCommand zcat, --quantMode TranscriptomeSAM GeneCounts, and --outSAMmapqUnique 0. The PRICE reference genome was prepared as described, utilizing hg19 FASTA and GTF files from Gencode. Next, PRICE was executed with the respective BAM files using the command -/Gedi/Gedi_1.0.5/gedi -e Price -D -genomic hg19 -progress -plot. Subsequently, all ORF features with a $P$ value of $P \leq 0.05$ were quantified using standard UNIX commands. ORF tables from PRICE were adjusted to the BED format, including chromosome, ORF feature start position, ORF feature end position, ORF feature ID, chromosome strand, and Gene ID. The resulting BED6 files were converted to BED12 format, which served as input for bedtools getfasta with the flags -s, -name, and -split. Peptide sequences were generated using the faTrans program.

## Translation efficiency

Translation efficiency of uORF/uoORFs was calculated by extracting IDs, start and end positions of predicted uORF and uoORF features ($P$ value <0.05) from PRICE ORF tables and arranged in SAF format, creating a uORF/uoORF SAF reference file. Next, read counts in uORF/uoORF regions were determined with featureCounts (v1.5.1) and genome-based BAM files from RNAseq and RiboSeq. The resulting aggregated count matrix was subjected to RiboDiff (v0.2.1) calculation. Features with missing calculations were discarded. Data were generated using pseudoreplicates to ensure robustness in translational efficiency estimates.

## Immunopeptidomics

Input cell lines (U-2 OS, SUM-159PT, and MDA-MB-231) for immunopeptidomics were harvested in triplicate with $5 \times 10^7$ cells per replicate. The cell lines U-2 OS, SUM-159PT, and MDA-MB-231 were treated with 1 μM Taxol for 16 h For DMSO-treated control conditions, cells were gently scraped in ice-cold PBS. For Taxol-treated conditions, mitotic shake-offs were spun at 600 × $g$ for 10 min at 4 °C and washed with ice-cold PBS. All samples of all conditions were counted using a CASY II system and snap-frozen.

Immunoprecipitation of HLA class I:peptide complexes was performed as described by ref. 59 with additional steps for the forced oxidation of methionine using $H_2O_2$ and reduction and alkylation of cysteine using tris(2-carboxyethyl)phosphine (TCEP) and iodoacetamide (IAA).

Lyophilized peptides were dissolved in 12 µl of 5% ACN in 0.1% TFA and spiked with 0.5 µl of 100 fmol/µl peptide retention time calibration (PRTC) Mixture (Pierce) and 10 fmol/µl JPTRT 11 (a subset of peptides from the Retention Time Standardization Kit; JPT) and transferred to QuanRecovery Vials with MaxPeak HPS (Waters, Milford, MA, USA). All samples were analyzed using an UltiMate 3000 RSLCnano system coupled to an Orbitrap Exploris 480 equipped with a FAIMS Pro Interface (Thermo Fisher Scientific). For chromatographic separation, peptides were first loaded onto a trapping cartridge (Acclaim PepMap 100 C18 µ-Precolumn, 5 µm, 300 µm i.d. × 5 mm, 100 Å; Thermo Fisher Scientific) and then eluted and separated using a nanoEase M/Z Peptide BEH C18 130 A 1.7 µm, 75 µm × 200 mm (Waters). Total analysis time was 120 min and separation was performed using a flow rate of 0.3 µl/min with a gradient starting from 1% solvent B (100% ACN, 0.1% TFA) and 99% solvent A (0.1% FA in $H_2O$) for 0.5 min. Concentration of solvent B was increased to 2.5% in 12.5 min, to 28.6% in 87 min and then to 38.7% in 1.4 min. Subsequently, the concentration of solvent B was increased to 80% in 2.6 min and kept at 80% solvent B for 5 min for washing. Finally, the column was re-equilibrated at 1% solvent B for 11 min. The LC system was coupled online to the mass spectrometer using a Nanospray-Flex ion source (Thermo Fisher Scientific), a SimpleLink Uno liquid junction (FossilIonTech) and a CoAnn ESI Emitter (Fused Silica 20 µm ID, 365 µm OD with orifice ID 10 µm; CoAnn Technologies). The mass spectrometer was operated in positive mode, and a spray voltage of 2400 V was applied for ionization with an ion transfer tube temperature of 275 °C. For ion mobility separation, the FAIMS module was operated with standard resolution and a total carrier gas flow of 4.0 l/min. Each sample was injected twice using either a compensation voltage of −50 or −65 V for maximal orthogonality and thus increased immunopeptidome coverage. Full Scan MS spectra were acquired for a range of 300–1650 m/z with a resolution of 120,000 (RF Lens 50%, AGC Target 300%). MS/MS spectra were acquired in data-independent mode using 44 previously determined dynamic mass windows optimized for HLA class I peptides with an overlap of 0.5 m/z. HCD collision energy was set to 28% and MS/MS spectra were recorded with a resolution of 30,000 (normalized AGC target 3000%).

FAIMS-DIA MS raw data were analyzed using Spectronaut software (version 17.6; Biognosys)[60] and searched against the UniProtKB/Swiss-Prot database (retrieved: 21.10.2021, 20,387 entries) as well as a database containing protein sequences longer than seven amino acids predicted from translation of uORFs in a single search. Search parameters were set to non-specific digestion and a peptide length of 7–15 amino acids. Carbamidomethylation of cysteine and oxidation of methionine were included as variable modifications. Identification results were reported with 1% FDR at the peptide level. Peptides identified by Spectronaut were further analyzed using NetMHCpan-4.1 binding predictions[61], Gibbs 2.0 clustering of peptide sequences[62], and retention time prediction by DeepLC[63]. uORF/uoORF-derived peptide sequences were manually validated using Skyline (version 22)[64,65] by comparison against spectral libraries in silico predicted using PROSIT[66]. Normalized spectral angles (NSAs) were calculated as described by ref. 67. Quantification of HLA class I-presented peptides was performed as described by ref. 68 using the raw output at the MS2 level from Spectronaut with cross-run normalization disabled. Peptides with a fold change in abundance >2 and with an FDR ≤0.05 were defined as "hits" while peptides with a fold change ≥1.5 and an FDR ≤0.2 were defined as "candidates".

Validation of uORF/uoORF-derived peptides was performed using parallel reaction monitoring (PRM)[43]. Briefly, top candidates identified by FAIMS-DIA were selected based on differential HLA presentation with a log2 fold change ≥0.2 and adjusted P value ≤0.05 and obtained as stable isotope labeled (SIL) synthetic peptides. PRM acquisition was performed using the following parameters: MS data were recorded with a resolution of 90,000 at 200 m/z, over the mass range from 202 to 1370 m/z, with 300% AGC target and 25 ms maximum IT. MS2 data were acquired with PRM scans using a resolution of 240,000 at 200 m/z. The target precursor list was provided with preselected charge states, the corresponding m/z and optimized CE values and a narrow isolation window (≤1 m/z) tuned per precursor, and their expected retention time (±1.5 min) predefined with SIL peptides and indexed to a set of retention time standard peptides. The normalized AGC target was set to 1000%. The maximum injection time mode was set to dynamic, aiming for a coverage of at least five points across the chromatographic peak. The dynamic RT feature using the PRTC mixture was active. Protonated polycyclodimethylsiloxane (a background ion originating from ambient air) at 445.12 m/z served as a lock mass. For spike-in experiments, 3 fmol of SILs were added, and additional MS2 data of SILs were acquired with a resolution of 45,000 at 200 m/z, normalized AGC target of 200% and a maximum injection time of 350 ms using the optimized CE values, narrow isolation windows and predefined and indexed expected retention time. Data were manually analyzed using Skyline[64,65] and R[43].

## Mice

OT-1 female mice[69] (6–8 weeks old) were bred in-house at our animal facility. All animals were housed under standardized laboratory conditions. To reduce environmental variation, experimental and control mice were co-housed. The facility maintained a 12-h light/dark cycle, with ambient temperature controlled at 22 ± 2 °C and relative humidity between 45 and 65%. Mice had ad libitum access to food and water and were housed in groups of 3 to 5 per cage. All animals were maintained in a specific pathogen-free (SPF) environment and handled in accordance with institutional and governmental regulations, as approved by the German Cancer Research Center and the regional authority (Regierungspräsidium Karlsruhe; G57-20), following the German Animal Protection Law and the European Directive 2010/63/EU on the protection of animals used for scientific purposes.

## CD8 + T cell isolation and culture

Primary naive CD8 + T OT-I cells were isolated using the MojoSort Mouse CD8 T cell isolation kit (BioLegend, 480007) and subsequently activated for 72 h on plates coated with 2 µg/ml anti-CD3 (BioXCell, BE0001-1) and 2 µg/ml anti-CD28 (BioXCell, BE0015-1) at 37 °C. The T cells were maintained in RPMI 1640 (Thermo Fisher Scientific, 21875-034) supplemented with 10% FBS, 1% penicillin-streptomycin, 1 mM sodium pyruvate (Thermo Fisher Scientific, 11360-070), 20 mM HEPES (Sigma, H4034), 50 µM β-mercaptoethanol (Sigma, M6250), and 10 ng/ml murine IL-2 (BioLegend, 575404).

## Quantitative real-time PCR

A total of 500 ng of RNA was reverse-transcribed using LunaScript RT Supermix (NEB, M3010L). Quantitative real-time PCR was subsequently performed with Luna Universal qPCR Mix (NEB, M3003X). Ct values were obtained using the QuantStudio 5 RT qPCR System and analyzed with QuantStudio Design and Analysis Software v2.6.0. mRNA fold change of target genes was calculated using the ΔΔCt method, with mRNA expression normalized to GAPDH. The primers used in this study are listed in Supplementary Data 13.

## 5' UTR cloning

A synthetic DNA template containing the 5' UTR and the coding sequence for the SIINFEKL peptide was amplified by PCR using primers with overlapping sequences complementary to the pGL3-Promoter vector (Promega). Following PCR, the fragments were gel-purified to remove any non-specific products. The pGL3-Promoter vector was linearized using HindIII-HF (NEB, catalog no. R3104). The purified insert and linearized vector were then combined with NEBuilder HiFi DNA Assembly Master Mix (NEB, catalog no. E2621) and incubated at 50 °C for 60 min. The assembled product was transformed into

competent cells for propagation and further analysis. Primers and synthetic DNA templates used in this study are listed in Supplementary Data 13.

## T cell killing assay
CD8[+] OT-I T cells were co-cultured with mouse cancer cells that express the specified uORF-SIINFEKL reporters at a ratio of 1:2. Following 24 h of incubation at 37 °C in a 5% $CO_2$ atmosphere, the cells were washed with PBS and stained with crystal violet to evaluate the killing efficiency. Imaging was conducted using the Dual Lens System V850 Pro Scanner (Epson), and colony area was quantified using the ImageJ plugin ColonyArea[70].

## Flow cytometry
TC1 cells were transfected with the uORF-SIINFEKL reporters using Lipofectamine 3000 (Thermo Fisher). Twenty-four hours post-transfection, cells were synchronized in mitosis by treatment with 1 μM Taxol (Santa Cruz, sc-201439) for an additional 16 h. Following mitotic arrest, cells were washed with PBS, detached using PBS-EDTA, and then pelleted. The cells were subsequently washed with PBS containing 0.5% BSA and incubated on ice and in the dark with APC-conjugated anti-mouse H-2Kb-SIINFEKL antibodies (Biolegend, clone 25-D1.16, #141606; 1:200) for 30 min. After incubation, the cells were washed twice with PBS containing 0.1% BSA and analyzed using a FACS Canto II cytometer (Thermo Fisher). Data analysis was conducted with FlowJo V10.4 software (FlowJo).

## IFN-γ quantification
Cytokine release from CD8 + T cells was measured from the cell supernatant using the ELISA MAX Deluxe Set Mouse IFN-γ (BioLegend, 430815), following the manufacturer's guidelines. Each sample was analyzed with the Multiskan FC plate reader (Thermo Fisher Scientific), using absorbance readings at 450 and 570 nm for subtraction. Final concentrations were calculated using a four-parameter logistic curve-fitting algorithm in GraphPad Prism.

## Peptides
Peptides used for ELISpot assays were synthesized by the Research Group GMP & T Cell Therapy of DKFZ, using Fmoc chemistry on a parallel peptide synthesizer with DIC and HBTU activation and Oxy-maPure as an additive. The crude peptides were purified to >95% purity by reversed-phase HPLC and characterized by analytical HPLC-MS. Lyophilized peptides were dissolved in DMSO at 10 mg/ml and stored at −80 °C.

## PBMC expansion
Experiments were performed with residual blood samples obtained from healthy platelet donors of the Hannover Medical School (MHH) Institute of Transfusion Medicine and Transplant Engineering of Hannover Medical School (Ethical vote number: 3639-2017). Sex/gender was not considered in the experimental design. PBMCs were extracted using discontinuous-gradient centrifugation. If available, samples from donors with exact HLA matches for the tested peptides were used. Otherwise, samples from donors with HLA alleles of the same supertype[71] were used.

For the expansion of epitope-specific T cells, the protocol published by ref. 72 was adapted. In short, PBMCs were thawed and washed in 10 mL X-Vivo15 medium (BE02-060F, Lonza) containing 2 μL Benzonase (Sigma Aldrich). Subsequently, PBMCs were resuspended in X-Vivo15 medium supplemented with 1000 IU/mL hrGM-CSF (R&D systems), 500 IU/mL IL-4 (R&D systems), and 50 ng/mL Flt3-L (R&D systems). For each peptide of interest, $1-2 \times 10^6$ cells were seeded in 2 mL per well in a 24-well plate. The next day, 1 mL medium was exchanged for 1 mL fresh X-vivo15 medium containing 200 ng/mL LPS (InvivoGen), 20 μM R848 (Sigma Aldrich), 20 ng/mL IL-1β (R&D

systems), and 10 μg/mL of the respective peptide of interest. A day later, the cells were fed by exchanging 1 mL medium per well for 1 mL T cell medium (RPMI1640 + 10% human serum + 100 μg/mL gentamicin + 2 mM L-glutamine + 10 mM HEPES) containing 20 IU/mL IL-2 (PeproTech), 20 ng/mL IL-7 (R&D systems) and 20 ng/mL IL-15 (PeproTech). This feeding was repeated every 2 to 3 days until day 7. On day 9, the feeding was performed with T cell medium only.

## ELISpot assays
At day 12, $0.5-3 \times 10^5$ cells/well of each T cell line were transferred to sterile Multiscreen-HA membrane plates (Millipore) coated with 1:500 anti-human IFNγ (clone 1-D1K; Mabtech) in PBS. Cell lines were re-stimulated with 10 μg/mL of the respective peptide (four replicates). Concanavalin A (Sigma Aldrich, 2 μg/mL) (one replicate) and DMSO (Sigma Aldrich, 1 μL/mL) (three replicates) were used as positive and negative controls, respectively. After 24 h of incubation, cells were discarded. Plates were washed with PBS and developed with 1:1000 biotinylated anti-human IFNγ (clone 7 B6-1-Biotin; Mabtech) in PBS, 1:2000 Streptavidin-Alkaline Phosphatase solution (Mabtech) in PBS, and filtered substrate (NBT/BCIP; Millipore). Plates were washed with ELISpot wash buffer (PBS with 0.05% Tween 20) between steps. Spots were counted with an automated ImmunoSpot reader (CTL-Immunospot® S6 Ultra-UV). For each well, the stimulation index (SI) was calculated by dividing the spot count by the mean spot count in DMSO-treated control wells. A stringent threshold of a stimulation index >3 was applied to define positive responses.

## Single-cell RNA sequencing
Data from single-cell RNA sequencing (scRNA-seq) corresponding to NCBI GEO Series GSE169246, including the count matrix, barcodes, feature annotations, and metadata, were downloaded (ref. 73). High-quality reads were aligned to the human reference genome GRCh38 using the Cell Ranger toolkit. The resulting matrix was filtered to remove cells with >10% mitochondrial gene content, fewer than 200 detected genes, fewer than 600 or more than 120,000 total counts, and genes expressed in fewer than ten cells. Potential doublets were identified and excluded using Scrublet and Scanpy. Metadata were then used to select barcodes corresponding to tumor-infiltrating CD4[+] or CD8[+] T cells from patients treated exclusively with Paclitaxel. For each condition, the frequency of CD4[+] or CD8[+] T cells was calculated by dividing the number of cells in each subset by the total T cell count. Frequencies were visualized with boxplots generated in R. Statistical significance between groups was assessed by two-sided Wilcoxon rank-sum tests.

## T cell phenotyping and intracellular cytokine staining
T cell lines that showed responses in ELISpot assays were split equally into two samples. One sample was re-stimulated with 10 μg/mL of its specific peptide, the other sample was mock-stimulated with 0.1% (v/v) DMSO. A T cell line only stimulated with DMSO was used as a control sample that was activated by treatment with 20 ng/ml Phorbol 12-myristate 13-acetate and 1 μM Ionomycin. GolgiStop (1:10, BD) and GolgiPlug (1:15, BD) were added immediately after the stimulation to all samples and the samples were incubated for 5 h at standard culture conditions. After incubation, the samples were washed with cell staining buffer (Biolegend) and then stained with antibodies against CD3 (OKT3, Biolegend), CD4 (RPA-T4, Biolegend), and CD8 (RPA-T8, BD), along with fixable live/dead NIR dye (Thermo Fisher) in 50 μL cell staining buffer at 4 °C for 30 min. Then, three washing steps with 200 μL cell staining buffer and centrifugation at $400 \times g$, 5 min were performed, followed by fixation with fix/perm solution (Cytofix/Cytoperm™ kit, BD) for 20 min at 4 °C. After fixation, the cells were again washed three times with 200 μL perm/wash buffer (Cytofix/Cytoperm™ kit, BD) and were then stained for IFNγ (4S.B3, Biolegend) and TNFα (Mab11, BD) in 50 μL perm/wash buffer at 4 °C for 30 min.

The samples were washed three times with perm/wash buffer and finally resuspended in 100 μL cell staining buffer. Samples were acquired at a BD FACS Canto™ II analyzer using the BD FACS Diva Software Version 6. The obtained data were analyzed with the FlowJo software Version 10.4. The gating strategy is depicted in Supplementary Fig. 8.

## Statistics and reproducibility

The number of biological replicates (*n*) is specified in each figure legend. No data were excluded from the analyses. The experiments were not randomized. The investigators were not blinded to allocation during experiments and outcome assessment. Statistical analyses were conducted using appropriate tests as indicated in the figure legends, including two-tailed unpaired *t*-tests, one-sided Fisher's exact tests, chi-square tests, Wilcoxon rank-sum tests, and empirical Bayes moderated *t*-tests. Adjusted *P* values were calculated using the Benjamini–Hochberg procedure where applicable. Data were presented as mean ± standard deviation (SD) unless otherwise stated.

## Reporting summary

Further information on research design is available in the Nature Portfolio Reporting Summary linked to this article.

## Data availability

The ribosome profiling and RNA-Seq data generated in this study are publicly available at the GEO repository under the accession code GSE281253. The mass spectrometry proteomics data have been deposited to the ProteomeXchange Consortium via the PRIDE partner repository with the dataset identifier PXD057839. Source data are provided with this paper.

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

## Acknowledgements

We thank Wilhelm Palm for advice and critical discussions. We thank Chong Sun for sharing reagents and technical advice. We thank Rebecca Köhler for excellent technical assistance. We thank Prof. Britta Eiz-Vesper, Hannover Medical School (MHH), Institute of Transfusion Medicine and Transplant Engineering, for providing HLA-typed PBMCs. This work was funded in part by grants of the European Research Council "DualRP" (ERC StG No. 759579) and the German Research Foundation (DFG 504774163 and DFG 545215964) to F.L.-P. R.A. is supported by the Dutch Cancer Society (KWF-13647), the European Research Council (Horizon ERC-2023-ADG-101141245), the Dutch Science Organization (OCENW.M.22.001-16221), and the AvL Foundation. A.K. and R.D.P. are supported by fellowships of the Helmholtz International Graduate School. Z.T. is supported by a scholarship from the China Scholarship Council (CSC).

## Author contributions

A.K., R.A., and F.L.-P. conceived the project, designed all the experiments, and wrote the manuscript. Methodology and data acquisition: A.K., J.P.B., R.D.P., Z.T., J.C., K.W., K.S., A.G.-A., P.-R.K., J.M.N., A.E., F.M.T., H.S., M.A.M.-P., Manuscript revision: A.K., J.P.B., A.B.R., R.A., and F.L.-P.

## Funding

## Competing interests

The authors declare no competing interests
