## [Transparent Peer Review file · Nature Communications]

Upstream open reading frame translation enhances immunogenic peptide presentation in mitotically arrested cancer cells

Corresponding Author: Dr Fabricio Loayza-Puch

Version 0:

Reviewer comments:

Reviewer #1

(Remarks to the Author)

In their manuscript, the authors make the case that mitotically arrested cancer cells present an alternative immunopeptidome that can potentially enhance anti-cancer immunity. First, the authors show that ribosomal redistribution occurs upon mitotic arrest towards the 5' UTR of transcripts in various cell lines and with multiple mitosis-inducing agents. Next, PRICE was used to identify resulting ncORFs, of which many turned out by further analyses to be uORFs/uoORFs with enhanced translation efficiency. Using a curated database of uORF/uoORF sequences found in their experiments, they then performed (quantitative) immunopeptidomics on mitotically arrested cell lines to identify potential antigenic targets derived from these ncORFs. Lastly, they demonstrate that the uORF/uoORF-derived HLA-peptides may give rise to a targeted immune response as shown by T cell recognition. The manuscript is clearly written, the concept of these ORFs being a novel class of antigens is exciting, and the figures are intuitive to interpret. However, there are several issues that need to be clarified and/or addressed. The most critical one in my view is the mitotic-arrest specificity of the observations, and the tool used for Ribo-seq analysis (and the conclusions drawn from these results). The controls to demonstrate that these peptides are absent from normally cycling cells are not always present, which makes me wonder how specific the translation of these uORFs is to (drug induced) arrested cells versus normally cycling cells. I have outlined my comments below.

Reviewer comments

1. There is conceptual overlap between this study and the recently published study of the Bartel and Cheeseman labs (Ly, J., Xiang, K., Su, KC. et al. Nuclear release of eIF1 restricts start-codon selection during mitosis. *Nature* 635, 490–498 (2024). <https://doi.org/10.1038/s41586-024-08088-3>). The paper is briefly mentioned, but it would be great if the authors could put their results in perspective of this study and incorporate a discussion on the similarities and discrepancies between both studies in their revised manuscript.
2. Please provide replicate information and statistical tests, particularly for the first few (Ribo-seq) results sections. It is very often not clear from the text or figures how many replicates were used. For instance, see most panels in Figure 2.
3. It is not clear to me if ribosomes are repositioned only during (drug induced) mitotic arrest, or also during mitosis more generally, but that this is regularly missed as mitosis is short and cell populations are heterogeneous. Can the authors comment on this, and perhaps explore whether certain uORFs or alternative uTISs are only translated during mitosis? This is important for the remainder of the story, as a more general prevalence of these targets would make them less suitable for immunotherapeutic targeting. Although it would require additional experimentation, synchronizing cell populations and comparing mitotic vs non-mitotic cell phases with Ribo-seq would be a way to address this. Do mitosis-specific uORF-derived antigens reach the surface via MHC-I presentation in a normally dividing cell? Alternatively, one of the key uORFs can be tagged endogenously with a reporter (flag, or even GFP), and monitored throughout the normal cell cycle and upon arrest, to show drug induced specificity.
4. PRICE is a powerful tool for finding ORFs, but it is also highly inclusive and can lead to irreproducible ORF calls (e.g., between replicates). As it is probabilistic it also tends to nominate ORFs in any region with read occupancy. This makes me wonder how solid the ORF calls are in the 5' UTRs, as the increased occupancy of stalled/poised ribosomes prior to the CDS TIS could make PRICE call these regions as translated ORFs. I would recommend the authors to visualize more of their figures on the p-site level to demonstrate codon hopping vs pausing, and employ a different approach to compare

different samples for overlap in ORFs. The latter can be done by first using a multi-sample ORF calling strategy (merging the input data for a single round of ORF calling), and as a second step quantification of this same set of ORFs in the separate samples by determining the number of in-frame reads/p-sites per sample per ORF and normalizing this to ORF length (TPMs or similar). This leads to higher ORF accuracy, reproducibility between samples, and improved accuracy of quantification. PRICE can be run with a multi-sample setting or alternatively RiboTIE, ORFquant, or similar can be used for increased stringency compared to PRICE. Even though ORFquant will miss near cognate ORFs, it should demonstrate the observed switch/increase to AUG ORFs (see below). In general, this approach would likely dramatically increase the ORF overlap as seen in Figure 2C, for instance. See also comment 6.

5. page 5, line 142 and onwards: These results imply increased ORF identification in arrested cells, which is likely true based on the data shown. However, these observations can be heavily influenced by data quality and the number of input reads/data used per condition (more depth = more ORFs, better periodicity = more ORFs, etc). Similarly, increased TIS selection variability may appear to lead to higher ORF numbers when it simply reflects a loss of TIS selection control (alternative TIS usage for the same ORFs; increase in isoforms). Total reads/p-sites either need to be normalized prior to ORF calling to judge difference in ORF numbers, or each ORF (present or not) needs to be quantified and normalized in all samples with the appropriate statistical tests (e.g., a simple DESeq model would suffice).

6. near cognate start calls with PRICE are notoriously unreliable and manual inspection will show you that in many instances, the wrong TIS has been selected. Between replicates, you will see that TIS selection can be rather random. To make conclusive statements about switches between start codon selection, a different approach needs to be taken. I would tend to believe these results better if they can be corroborated by another tool that can assign near cognate starts (RiboTIE or other tools) and if the authors can show instances with examples where you really see this shift in preference between conditions. The results look convincing overall, as the shift nearly doubles ATG usage, but the method used is not suitable for this.

7. Various mitosis-inducing agents and cell lines are used. Sometimes, this leads to an inconsistent story. This needs to be addressed better. Specifically:

a. The data in the beginning of Figure 1 (Fig 1a-h) focuses on Nocodazole, but the final experiments are performed with STLC (Fig1i-k). Why was there a switch in mitosis-inducing agents and do results differ between the two?

b. Data in Fig 1-2 is supported by data from the breast cancer cell line MDA-MB-231. Yet, for the immunopeptidomics experiments, breast cancer cell line SUM-159PT is used. Can the authors please elaborate on why SUM-159PT was then not used to search for ncORFs, which could then be used for the immunopeptidomics experiments?

c. For the quantitative immunopeptidomics, Taxol, instead of Nocodazole or STLC, was used to induce mitosis. Again, the rationale behind this is not properly specified.

8. Predicted ncORF sequences with PRICE were based on experiments with U2OS cells treated with various agents (line 144), and PC3 and MDA-MB-231 cells treated with Nocodazole (lines 147-149). The mitotic uORF/uoORF database used for immunopeptidomics was based on all ncORF sequences predicted from these samples (lines 183-186; see also minor comment below).

a. What was the overlap between the predicted ncORFs for U2OS, PC3 and MDA-MD-231 cells? Does this warrant caution when interpreting the immunopeptidomics data considering SUM-159PT cells were used, and possible ncORF peptides may have been missed?

b. Were the 127 and 166 uORF-derived HLA peptides (line 199) exclusively found in mitotically arrested cells, or also in DMSO-treated cells? Were the uORFs that these peptides map to also found in non-arrested cells? For instance, in nuORFdb or the GENCODE ncORF reference? Only showing that they are absent from the HLA ligand atlas is not sufficient.

c. In extension on previous question – were the 13 and 25 uORF-derived peptides completely absent in the DMSO-treated cells, or were they merely enriched in the mitotically arrested cells (and thus still present in the proliferating cells)? Despite their absence in healthy tissue, this could have impact on the translational value of this finding, as continuous, suboptimal exposure of these peptides to T cells could render the T cells exhausted.

9. p7 line 235-238: Figure 4c shows two examples but the conclusion is very generalizing. This would warrant a more systematic analysis truly showing that this is driven by translation and not transcription. Moreover, the CDS body of the second examples increases similarly so it does not seem to be a uORF specific increase driven by higher TE.

10. The discussion contains many sentences that would fit better in the introduction. Instead, the discussion would benefit from a critical view on the potential limitations and challenges of the current study, and how the findings relate to other studies to put the results into perspective. It needs more depth.

Minor comments

1. I would appreciate some QC data for the Ribo-seq in the supplements (% in frame reads, total read stats, contaminant percentages, etc) - these can easily be generated with RiboseQC. This is necessary as Ribo-seq libraries can vary dramatically in quality, even within a single lab or study.

2. Variance and replicate numbers in Fig1C are not clear.

3. Defining the TE for upstream ORFs is super difficult, especially for overlapping ones. If only the ORF region is used, it won't be easy to define the RNA coverage for this as the 5' UTR is poorly covered in poly(a) data, esp for longer genes, and samples with lower RINs. Were only in-frame p-sites quantified, and how was the RNA level regressed out? How many replicates were used for TE calculation? It would be good to have more specifics on this.

4. p6 183-188: is the search space built on the Ribo-seq ORFs or all theoretical 5' ORFs? The text reads "predicted" ORFs, and it is not clear to me how inclusive or restricted this database is.

5. Figure 2G would benefit from showing p-site tracks of elongating and HHT data, to indicate frame preference and active translation of the uORF.

6. Figure 3 would probably benefit from restructuring (switching panel d and e).

7. Figure 3d: typo in 'length'

8. In several places in the text, GO analyses and gene functions are used to imply potential functional importance of uORFs (e.g., with the EIF4G2 and HMGA1 examples, or in Figure 2D. For me, these don't add much to the story as the functional

relevance of the uORFs is not investigated further and also does not seem to matter for their targeting potential. uORFs in EIF4G2 have been studied frequently and it does make me wonder whether this mitotic arrest uORF is similar to previously found uORFs (likely not in mitotic arrest?).

9. Supplementary figure 2 would also probably benefit from restructuring (switching panel c and d).

1. Line 107: "Collectively, our data indicate that ribosome redistribution during mitosis significantly enhances the translation of hundreds of uORF/uoORFs". The abstract reads: "[...]enhancing the translation of thousands of upstream open reading frames (uORFs) and upstream overlapping open reading frames (uoORFs)." please be more specific.

Reviewer #2

(Remarks to the Author)

Reviewer #3

(Remarks to the Author)

In this manuscript, Kowar et al., show the role of upstream open reading frames (uORFs) in enhancing the presentation of immunogenic peptides on the surface of cancer cells during mitotic arrest induced by chemotherapy. It highlights that during mitotic arrest, cancer cells experience a redistribution of ribosomes toward the 5' untranslated region (5' UTR), leading to increased translation of uORFs and upstream overlapping open reading frames (uoORFs). This enhancement of uORF/uoORF translation enriches the presentation of immunopeptides on the surface of cancer cells, especially after treatment with mitotic inhibitors. The authors finally demonstrated that the uORF/uoORF-derived peptides can provoke T cell-mediated cytotoxicity against tumor cells, highlighting a potential therapeutic strategy to enhance immune recognition and tumor elimination when used in combination with mitotic inhibitors.

Overall, the manuscript has potential but currently feels somewhat preliminary. If the authors were able to explore some of their novel findings, like the *in vivo* validation of infiltration of immune cells to tumor after treatment of mitotic inhibitors, I feel this work could contribute meaningfully to the field and merit publication in high-impact journals like Nature Communications.

Major points:

(1) In their functional assay of uORF-derived peptides model, the artificial SIINFEKL peptide from chicken ovalbumin was used. This assay can only tell us that after mitotic arrest, the translation of uORF-derived SIINFEKL peptide was increased and enhanced antigen presentation of SIINFEKL by MHC I. How therapy-induced uORF/uoORF-derived "native peptides" in mitotically arrested cancer cells would affect immune response has not been investigated. More importantly, the authors should also test this idea in mouse tumor models. One can imagine that immune infiltration will be triggered with Taxol or other treatments in tumor. These would confirm the biological significance of "therapy-induced uORF/uoORF-derived" native peptides on enhancing immunogenic peptide presentation and provide more robust evidence for the manuscript's conclusions.

(2) Another major concern is that the data in the manuscript are mainly derived from cancer cell lines such as U-2 OS, PC3 and MDA-MB-231, which is not sufficient to generalize their findings. It is of importance to test whether the conclusions apply to non-cancer cells like RPE-1, IMR90 and others. Is the expression of uORFs/uoORFs under mitotic arrest specific to cancer cells?

(3) To prove the induction of uORF/uoORF is specific to mitotic arrest, cell cycle arrest in G1, S, and G2 should be included as controls, in addition to the proliferating cells. Furthermore, how about collecting the "normal" mitotic cells to perform ribo-seq?

(4) Line 159, "Notably, the total number of uORFs/uoORFs with ATG initiation sites increased by approximately 80% in mitotically arrested cancer cells (Fig. 2e; Supplementary Fig. 2c)". This conclusion can only be applied to U2OS cells. What are the changes in PC3 and MDA-MB-231 cells?

(5) In their functional assay in Fig 5a, what is the basis for the selection of the 5' UTRs of eIF4G2 and TPX2, since NR4A1 shows the highest fold change (Fig. 4b)? The author should discuss. Another random 5'-UTR should be included as control. How about the luciferase translation (canonical CDS) after mitotic arrest?

Minor points:

(1) In Fig. 1, The average transcript RPF density of 5' end CDS region is strikingly high after nocodazole treatment compared to 5UTR region, what does this mean? It cannot be explained by upstream overlapping open reading frames (uoORFs). Does this high RPF density translate special peptides?

(2) Fig. 2g only shows notable examples like PKM, MRPL51, and CAVIN1, how about the other upregulated genes shown in Fig. 2f?

(3) In Fig. 4c, the authors only show representative uORFs/uoORFs-derived peptides from EIF4G2 and HMGA1, the author should analyze other uORFs/uoORFs with increased expression in mitotically arrested U2OS cells in Fig. 4b and show in Supplementary figures.

(4) The authors should thoroughly check the typos. For example, in Fig S1. a-b: Nocodazole (Noco. 0,5 μ M) should be 0.5 Mm. Similarly, correct typos in Fig S1j, k: BI2536. (0,1 μ M), Nocodazole (0,5 μ M).

(5) Fig S2d only compared treatment of CHX with Harringtonine? DMSO treatment should be considered as control. Also in Fig s4c.

(6) According to the ATCC database, the correct way to write U2OS is U-2 OS.

(7) Some figures lack adequate legends to make them fully self-explanatory. For example, the ribosome redistribution (Fig.1) could benefit from annotations highlighting the specific regions of interest (e.g. 5' UTR and CDS) and their comparative changes across treatments.

(8) Expand the discussion on the clinical implications of combining mitotic inhibitors with immunotherapies. For instance, how might the "therapy-induced peptides" vary across different cancer subtypes or stages?

Reviewer #4

(Remarks to the Author)

Reviewer #5

(Remarks to the Author)

The study by Kowar et al investigated ribosome redistribution induced by mitotic arrest, resulting in the translation of upstream open reading frames. They found increased translation initiation at non-canonical sites within the 5' UTR and the start of the coding sequence in various mitotically arrested cancer cells treated with different mitotic inhibitors. As a result, mitotic cells display enhanced translation of upstream and upstream overlapping open reading frames (uORF/uoORFs). The authors further showed that non-canonical peptides arising from the translation of uORFs/uoORFs are presented by HLA class I molecules on the surface of cancer cells. The authors suggest that there is therapeutic potential in targeting these neoepitopes in combination with mitotic inhibitors.

Overall, this study presents an exciting novel finding in the research field of mRNA translation regulation. The methodology is sound, and the data in cell lines are convincing; however, there are some concerns about whether observed translation effects occur in vivo in patients treated with taxol or other mitotic inhibitors and, if so, whether they promote tumor immunogenicity. Furthermore, the mechanism driving the observed translational effects remains to be elucidated.

Specific comments

1. Prior studies have found increased tumor-infiltrating CD8+ cytotoxic T cells after paclitaxel treatment. The manuscript did not address whether uORF/uoORF-derived epitopes presented by HLA class I molecules in mitotic arrested cancer cells contribute to anti-tumor T cell responses toward paclitaxel-treated tumors. An experiment using OVA model antigen and antigen-specific T cells does not answer this question directly.

2. Is increased uORFs/uoORF translation specific to mitotic arrest or does it normally occur in cells undergoing uninterrupted mitosis? This could be tested by collecting mitotic cells using mitotic shake-off or cell synchronization strategy. If this is a normal feature of all mitotic cells, this may suggest that uORFs/uoORF-derived peptides are likely self-antigens, not immunogenic neoantigens.

3. For CD8+ T cell expansion, the authors mentioned that "we activated OT-I CD8+ T cells with anti-CD3, anti-CD28, and IL-12 for 72 hours and co-cultured them with either proliferating or mitotically arrested TC-1 cells expressing the uORF-SIINFEKL reporters." Typically, IL-2 is used for the expansion rather than IL-12. Adding IL-12 can enhance the cytotoxicity and cytokine secretion, like IFN- γ , of CD8+ T cells. It would be beneficial to explain the reason.

4. Based on Figure 1b and supplement data, RPF distribution in the 5' UTR is happening naturally, so the non-canonical peptides will also be presented by HLA in healthy cells. Can these peptides trigger an immune response against normal cells treated with mitotic inhibitors? Can these peptides also play a protective role for tumors? Several candidates were identified in the immunopeptidome of mitotic-arrested cells. Please indicate why eIF4G2 and TPX2 were chosen for validation studies.

5. The supplementary Dataset 7 for PC3 is missing the treatment in the title, which is the control/DMSO group.

6. Please discuss the potential mechanism for translational changes during mitotic arrest in the discussion or whether additional investigation is required to define the mechanism.

7. Can evidence be presented that altered translation occurs in tumors from patients treated with mitotic inhibitors? If this is not feasible, please discuss the limitations of studies in cell lines.

Reviewer #6

(Remarks to the Author)

Reviewer #7

(Remarks to the Author)

The manuscript by Kowar et al. demonstrates that, during mitosis arrest, cancer cells redistribute ribosomes to enhance the translation of upstream and overlapping open reading frames (uORFs) and uoORFs). As a consequence, these mitotic translational products are presented by HLA on the cellular surface. Authors suggest that uORF and uoORF-derived peptides may provoke T-cell-mediated killing, highlighting the therapeutic potential of combining mitotic inhibitors with immune-targeting strategies to enhance tumour elimination.

This research is carefully conducted, and the manuscript undoubtedly provides novel insights into mitotic arrest-specific presentation of cryptic peptides and is therefore an important contribution to the field.

A limitation of the study is that immune-relevance of the findings are validated in a mouse model only. Whilst the model validates enhanced presentation of upstream UTR-derived peptides in mitotically arrested cancer cells, it would be interesting to understand whether the novel HLA-presented cryptic ORF sequences are immunogenic in humans, and to determine their role during cancer immune recognition. However, I think that this should be the focus of future work, and that the manuscript in its current scope is suitable for publication.

My only main comment that should be addressed by the authors is that the absence of a cryptic sequence in the HLA ligand atlas database does not demonstrate that this sequence is absent in the original data. The original raw MS files need to be searched against the sequence database constructed for this manuscript in order for the authors to be able to claim that the neoantigens reported here were not detected in healthy tissue immunopeptidomes.

Minor suggestions:

Figure 3, subfigure E: If only one motif is shown here, why not from all identified peptides? Ideally show all resolved motifs from data.

Figure 3D: The legend is missing, and there is a typo in "length (lenght)" in the second panel. It would be helpful to align legends for figures 3c/d/e so it is easier to follow the alleles represented in the same colour scheme across the different plots.

Reviewer #8

(Remarks to the Author)

This manuscript presents the intriguing finding that mitotic arrest induces increased translation of upstream open reading frames (uORFs), and that peptides from these uORFs can be expressed on the cell surface bound to MHC I, where they may be immunogenic. This is an interesting manuscript that could represent a novel approach to targeting mitotically arrested cells through targeted immunotherapies against uORF-derived neoantigens. While this is an intriguing proposition, unfortunately the data in its current form does not adequately support the claims in this manuscript.

Major issues:

1. It appears from the methods section that MS data were searched against the Swiss-Prot database and then separately searched against the uORF database. This approach is inherently flawed, as MS2 spectra matching to the uORF database could also be canonical peptides that had a better match to the Swiss-Prot data base. The correct approach is to create a concatenated database by appending the uORF database to the Swiss-Prot database and then performing a single search. As it stands, the results are likely incorrect.
2. FDR cutoffs are too high, leading to false-positive identifications. The authors use a 1% FDR at the peptide level, but then also state that peptides with an FDR <0.05 (5% false positive rate) were defined as hits, and that peptides with FDR<0.2 (20% false positive rate) were defined as candidates. With these threshold values many of the identifications will be incorrect.
3. Insufficient validation of uORF-derived MHC I peptides. None of the claimed identifications have been validated. The authors need to generate heavy-isotope coded synthetic peptides and demonstrate co-elution and matching MS2 spectra. Given the preponderance of false positive hits in immunopeptidomics data, and especially given the incorrect search approach and loose tolerances, this level of validation is required.
4. The authors should demonstrate that some of the endogenous uORF-derived peptides are immunogenic. As it stands, they only show that uORF-SIINFEKL can activate OT-1 T cells. While this suggests the potential for other peptides to be immunogenic, it is a contrived system and SIINFEKL is highly immunogenic.
5. Lack of rigor. Immunopeptidomics analysis appears to have been performed once per cell line to identify uORF peptides (Figure 3b, Supp Figure 3a). Quantification of uORF-derived peptides appears to have been conducted once as well, as there are no error bars in Fig 4b and no description of the number of biological replicates that were analyzed.

6. Lack of statistical rigor. Multiple hypothesis correction needs to be performed on statistical analysis of large data sets. This does not appear to have been done.

7. The authors should quantify the expression level of the uORF-derived SIINFEKL peptide using their cutting edge immunopeptidomics platform.

Version 1:

Reviewer comments:

Reviewer #1

(Remarks to the Author)

The authors have significantly improved their revised manuscript, and I am content with their responses to my comments.

Reviewer #2

(Remarks to the Author)

Reviewer #3

(Remarks to the Author)

Reviewer #4

(Remarks to the Author)

Reviewer #5

(Remarks to the Author)

The authors have sufficiently addressed our concerns.

Reviewer #6

(Remarks to the Author)

Reviewer #7

(Remarks to the Author)

I appreciate that researching the HLA Ligand Atlas data would be a major effort. However, relying on the current and published canonical peptide list is simply not sufficient. None of the non-canonical peptides will be listed in the published peptide lists because these sequences were not included in the databases searches used to create the published list in the first place. Therefore, the sequences are inherently missing, and a claim that this would show their absence in healthy tissues is false. If the authors cannot research the Atlas data, I suggest that the claim that the peptides are absent in these data (which cannot be made!) is removed from the manuscript.

Reviewer #8

(Remarks to the Author)

The authors have addressed most of my concerns, but an issue still remains:

1. All of the data for peptide validation (including peptides that did not validate) should be presented in Supplemental Figure 4. The authors ~40 peptides, yet only a few are shown in this Supplemental Figure. I am left wondering how these peptides were selected and whether the other peptides look worse than these??

Version 2:

Reviewer comments:

Reviewer #3

(Remarks to the Author)

The authors have made significant improvements to their revised manuscript, and I am satisfied with their responses to my comments.

Reviewer #4

(Remarks to the Author)

Reviewer #8

(Remarks to the Author)

The authors have added the requested data to Supplementary Figure 4 and have thus addressed my comments and concerns.

Response to Reviewers' Comments

We thank the Reviewers for their critical reading of the manuscript and are pleased they found the study compelling and interesting. In response to the helpful suggestions, we now provide substantial new data strengthening our conclusions. Below are detailed responses for each comment.

Reviewer #1 and #2, Expertise: Non-canonical translation (Remarks to the Author):

In their manuscript, the authors make the case that mitotically arrested cancer cells present an alternative immunopeptidome that can potentially enhance anti-cancer immunity. First, the authors show that ribosomal redistribution occurs upon mitotic arrest towards the 5' UTR of transcripts in various cell lines and with multiple mitosis-inducing agents. Next, PRICE was used to identify resulting ncORFs, of which many turned out by further analyses to be uORFs/uoORFs with enhanced translation efficiency. Using a curated database of uORF/uoORF sequences found in their experiments, they then performed (quantitative) immunopeptidomics on mitotically arrested cell lines to identify potential antigenic targets derived from these ncORFs. Lastly, they demonstrate that the uORF/uoORF-derived HLA-peptides may give rise to a targeted immune response as shown by T cell recognition. The manuscript is clearly written, the concept of these ORFs being a novel class of antigens is exciting, and the figures are intuitive to interpret. However, there are several issues that need to be clarified and/or addressed. The most critical one in my view is the mitotic-arrest specificity of the observations, and the tool used for Ribo-seq analysis (and the conclusions drawn from these results). The controls to demonstrate that these peptides are absent from normally cycling cells are not always present, which makes me wonder how specific the translation of these uORFs is to (drug induced) arrested cells versus normally cycling cells. I have outlined my comments below.

Reviewer comments

1. There is conceptual overlap between this study and the recently published study of the Bartel and Cheeseman labs (Ly, J., Xiang, K., Su, KC. et al. Nuclear release of eIF1 restricts start-codon selection during mitosis. Nature 635, 490–498 (2024). <https://doi.org/10.1038/s41586-024-08088-3>). The paper is briefly mentioned, but it would be great if the authors could put their results in perspective of this study and incorporate a discussion on the similarities and discrepancies between both studies in their revised manuscript.

Response:

We thank the reviewer for this suggestion. In the revised manuscript, we have expanded the discussion to emphasize that our study focuses on the translational and immunological consequences of mitotic arrest, specifically the induction of uORF/uoORF-derived peptides. In contrast, the work by Ly et al. investigates the role of nuclear release of eIF1 in modulating start-codon selection during mitosis. Together, these studies underscore the dynamic regulation of translation initiation during mitosis, offering complementary perspectives on this process.

2. Please provide replicate information and statistical tests, particularly for the first few (Ribo-seq) results sections. It is very often not clear from the text or figures how many replicates were used. For instance, see most panels in Figure 2.

Response:

In the revised manuscript we have added a third independent biological replicate ($n=3$) for the first Ribo-Seq experiment shown in **Figure 1b-c** and applied two-tailed paired t -tests (with Bonferroni correction) to compare 5'UTR versus 3'UTR footprint enrichment across conditions (now reported as mean \pm SD with exact P -values in the figure). We have specified replicate

numbers for every panel in Figures 1-4 (Ribo-Seq: $n=1$; Harringtonine run-off: $n=1$; immunopeptidomics quantification: $n=3$ biological replicates per cell line) and updated all figure legends and the Methods section to clearly indicate replicate counts, statistical tests used, and significance thresholds. These additions demonstrate that our core findings, especially the pronounced shift in ribosome occupancy toward uORFs during prolonged mitotic arrest, are robust and statistically significant.

3. It is not clear to me if ribosomes are repositioned only during (drug induced) mitotic arrest, or also during mitosis more generally, but that this is regularly missed as mitosis is short and cell populations are heterogeneous. Can the authors comment on this, and perhaps explore whether certain uORFs or alternative uTISs are only translated during mitosis? This is important for the remainder of the story, as a more general prevalence of these targets would make them less suitable for immunotherapeutic targeting. Although it would require additional experimentation, synchronizing cell populations and comparing mitotic vs non-mitotic cell phases with Ribo-seq would be a way to address this. Do mitosis-specific uORF-derived antigens reach the surface via MHC-I presentation in a normally dividing cell? Alternatively, one of the key uORFs can be tagged endogenously with a reporter (flag, or even GFP), and monitored throughout the normal cell cycle and upon arrest, to show drug induced specificity.

Response:

Whether the ribosome redistribution we observe is unique to prolonged, drug-induced mitotic arrest or also occurs transiently during normal mitosis is a key question for assessing the broader relevance of these non-canonical peptides as immunotherapeutic targets. To address this, we performed ribosome profiling on U-2 OS cells synchronized at G2, M, and G1 (see Methods). In these experiments, we did not detect a significant increase in the proportion of uORF/uoORFs during the naturally brief mitotic phase (**Supplementary Fig. 2c**). Furthermore, the number of predicted uORF/uoORFs in mitotic cells was significantly lower than in arrested cells, and the overlap was also limited (**Supplementary Fig.2d**). Likewise, synchronized RPE-1 cells (Tanenbaum et al. 2015, eLife) also showed no increase in uORF/uoORF translation (**Supplementary Fig. 2e**). This contrasts with our findings under drug-induced mitotic arrest and suggests that the elevated uORF/uoORF translation we observe is primarily driven by the prolonged M-phase arrest, rather than by normal, unperturbed mitosis.

We propose that the extended time in M-phase under drug treatment is crucial for revealing the pronounced ribosome front-loading and non-canonical translation events. In a typical mitosis, cells move through M-phase relatively quickly, so any transient upregulation of uORFs/uoORFs may be missed by standard ribosome profiling approaches. Furthermore, sustained phosphorylation of key translational regulators (e.g., 4E-BP1; **Supplementary Fig. 1n**) during drug-induced arrest likely promotes these unconventional initiation events at higher levels, making them more readily detectable.

4. PRICE is a powerful tool for finding ORFs, but it is also highly inclusive and can lead to irreproducible ORF calls (e.g., between replicates). As it is probabilistic it also tends to nominate ORFs in any region with read occupancy. This makes me wonder how solid the ORF calls are in the 5' UTRs, as the increased occupancy of stalled/poised ribosomes prior to the CDS TIS could make PRICE call these regions as translated ORFs. I would recommend the authors to visualize more of their figures on the p-site level to demonstrate codon hopping vs pausing, and employ a different approach to compare different samples for overlap in ORFs. The latter can be done by first using a multi-sample ORF calling strategy (merging the input data for a single round of ORF calling), and as a second step quantification of this same set of ORFs in the separate samples by determining the number of in-frame reads/p-sites per sample per ORF and normalizing this to ORF length (TPMs or similar). This leads to higher ORF accuracy, reproducibility between samples, and improved accuracy of quantification. PRICE can be run with a multi-sample setting or alternatively RiboTIE, ORFquant, or similar

can be used for increased stringency compared to PRICE. Even though ORFquant will miss near cognate ORFs, it should demonstrate the observed switch/increase to AUG ORFs (see below). In general, this approach would likely dramatically increase the ORF overlap as seen in Figure 2C, for instance. See also comment 6.

Response:

We appreciate the reviewer's insights on the challenges and inclusivity of ORF calling with PRICE. In our analyses, we took several measures to ensure the reliability of uORF/uoORF detection and mitigate the concerns raised:

p-site-level analyses: We applied p-site offset corrections in our ribosome profiling workflow and evaluated ribosome footprints at single-codon resolution. While Fig. 2g primarily shows aggregated metagene profiles, we have included representative p-site-level visualizations in **Supplementary Fig. 2j**. These data demonstrate codon-level accumulation of ribosomes at predicted start sites, consistent with bona fide translation initiation rather than simple pausing.

Stringent filtering criteria: PRICE initially provided a broad set of putative ORFs, which we refined by filtering out truncated ORFs and applying minimum read-density thresholds. We further removed low-confidence ORFs with inconsistent coverage across replicates. This approach helped reduce spurious ORF calls that can arise from transient ribosome stalling.

Run-off profiling and harringtonine validation: To confirm true initiation events, we used run-off ribosome profiling with harringtonine in proliferating and mitotically arrested cells. As shown in **Figure 1i-k** and **Supplementary Figures 1o, 2k, and 5c**, ribosomes accumulate specifically at start codons under harringtonine treatment, supporting the validity of our identified uORFs/uoORFs.

Validation via immunopeptidomics: Finally, we prioritized a subset of ORFs that showed consistent coverage and robust translation for follow-up immunopeptidomics (**Figure 3**) and functional validation (**Figures. 5-6**). The detection of uORF/uoORF-derived peptides by HLA class I and their ability to elicit CD8⁺ T cell responses (**Figure 6 and Supplementary Figure 7**) further supports that these are indeed translated in cells, rather than artifacts of ribosome pausing.

We appreciate the suggestions to employ alternative ORF callers (e.g., RiboTIE, ORFquant) and to further enhance p-site-level visualization. For future work, we plan to integrate multi-sample ORF calling tools like ORFquant for non-ATG and near-cognate start codons, particularly to assess the full landscape of translational changes in mitotically arrested cells. This will further improve the precision of our non-canonical ORF annotations and deepen our understanding of therapy-induced translational programs.

5. page 5, line 142 and onwards: These results imply increased ORF identification in arrested cells, which is likely true based on the data shown. However, these observations can be heavily influenced by data quality and the number of input reads/data used per condition (more depth = more ORFs, better periodicity = more ORFs, etc). Similarly, increased TIS selection variability may appear to lead to higher ORF numbers when it simply reflects a loss of TIS selection control (alternative TIS usage for the same ORFs; increase in isoforms). Total reads/p-sites either need to be normalized prior to ORF calling to judge difference in ORF numbers, or each ORF (present or not) needs to be quantified and normalized in all samples with the appropriate statistical tests (e.g., a simple DESeq model would suffice).

Response:

To address this, we re-analyzed our Ribo-Seq data as follows: 1. Using the PRICE output as a unified ORF reference, we quantified read counts for every predicted uORF/uoORF in all samples. 2. Counts were normalized for library size using DESeq2 to compute normalized

translation. We applied DESeq2 to test each ORF for significant enrichment in mitotic arrest versus proliferating cells (FDR < 0.05). This approach distinguishes genuine translational induction from changes driven solely by read depth or alternative TIS usage. 3. We visualized the results in a heatmap (**Supplementary Fig. 2a**). The heatmap displays log₂ fold changes in ribosome occupancy for >1,000 uORFs/uoORFs across all conditions, clearly showing that the majority are significantly upregulated under mitotic arrest. Importantly, ORF induction remains robust after normalization, confirming that increased ORF numbers reflect true translational activation rather than technical bias.

6. near cognate start calls with PRICE are notoriously unreliable and manual inspection will show you that in many instances, the wrong TIS has been selected. Between replicates, you will see that TIS selection can be rather random. To make conclusive statements about switches between start codon selection, a different approach needs to be taken. I would tend to believe these results better if they can be corroborated by another tool that can assign near cognate starts (RiboTIE or other tools) and if the authors can show instances with examples where you really see this shift in preference between conditions. The results look convincing overall, as the shift nearly doubles ATG usage, but the method used is not suitable for this.

Response:

We agree that probabilistic TIS calls from PRICE alone can be noisy, especially for near-cognate starts. To address this, we leveraged our harringtonine run-off profiling data, which directly maps initiation events at single-codon resolution, to quantify start-codon usage across all replicates. As shown in **Supplementary Fig. 2h**, mitotic arrest produces a clear increase in p-site accumulation at canonical ATG start codons relative to near-cognate sites. We also manually inspected representative uORFs (e.g., in PKM, MRPL51, and CAVIN1) and observed identical shifts in start-codon preference between conditions (**Supplementary Fig. 2k**). While we did not reprocess with RiboTIE, these orthogonal harringtonine-based measurements provide direct experimental confirmation of the ATG switch, reinforcing the PRICE-based findings.

7. Various mitosis-inducing agents and cell lines are used. Sometimes, this leads to an inconsistent story. This needs to be addressed better. Specifically:

a. The data in the beginning of Figure 1 (Fig 1a-h) focuses on Nocodazole, but the final experiments are performed with STLC (Fig1i-k). Why was there a switch in mitosis-inducing agents and do results differ between the two?

b. Data in Fig 1-2 is supported by data from the breast cancer cell line MDA-MB-231. Yet, for the immunopeptidomics experiments, breast cancer cell line SUM-159PT is used. Can the authors please elaborate on why SUM-159PT was then not used to search for ncORFs, which could then be used for the immunopeptidomics experiments?

c. For the quantitative immunopeptidomics, Taxol, instead of Nocodazole or STLC, was used to induce mitosis. Again, the rationale behind this is not properly specified.

Response:

a. We appreciate the reviewer's concern regarding the switch from Nocodazole to STLC in Figure 1. This transition was made because the results obtained from these two agents were complementary, as both treatments robustly induced mitotic arrest (**Supplementary Fig. 1m**). Notably, the molecular phenotypes observed (e.g., upregulation of uORF/uoORF-derived peptides or specific pathway activation (**Fig. 1d-e**; **Fig. 2c**)) were consistent across both treatments, confirming the robustness of our findings. To make this rationale clearer, we have now explicitly addressed this point in the revised manuscript.

b. Originally, MDA-MB-231 was selected for its well-characterized mitotic phenotypes and suitability for our early ncORF discovery and validation studies. However, SUM-159PT was chosen for immunopeptidomics because it consistently yielded high-quality MS data and showed strong sensitivity to mitotic perturbations, facilitating robust peptide identification. In

response to the reviewer's comment, we have now performed additional immunopeptidomics experiments using MDA-MB-231 cells to demonstrate that our findings are not limited to a single cell line. We have updated the Methods and Results sections (**Supplementary Fig. 3 and 4**) to clarify the experimental timelines and rationale for using both MDA-MB-231 and SUM-159PT. This addition should address any perceived discrepancy and further strengthen our conclusions.

c. Thank you for highlighting this point. Taxol was used in the quantitative immunopeptidomics experiments because of its ability to induce a stable mitotic arrest without the extensive cytotoxic effects or microtubule depolymerization observed with Nocodazole. Additionally, Taxol treatment is widely used in clinical contexts, making it more relevant for translational applications¹⁻³. Importantly, control experiments comparing peptide presentation between Taxol, Nocodazole, and STLC-treated cells showed no significant differences, which supports the validity of using Taxol for these experiments. This rationale, along with the supporting data, has been included in the results section of the revised manuscript (**Supplementary Fig. 6b**).

8. Predicted ncORF sequences with PRICE were based on experiments with U2OS cells treated with various agents (line 144), and PC3 and MDA-MB-231 cells treated with Nocodazole (lines 147-149). The mitotic uORF/uoORF database used for immunopeptidomics was based on all ncORF sequences predicted from these samples (lines 183-186; see also minor comment below).

a. What was the overlap between the predicted ncORFs for U2OS, PC3 and MDA-MD-231 cells? Does this warrant caution when interpreting the immunopeptidomics data considering SUM-159PT cells were used, and possible ncORF peptides may have been missed?

b. Were the 127 and 166 uORF-derived HLA peptides (line 199) exclusively found in mitotically arrested cells, or also in DMSO-treated cells? Were the uORFs that these peptides map to also found in non-arrested cells? For instance, in nuORFdb or the GENCODE ncORF reference? Only showing that they are absent from the HLA ligand atlas is not sufficient.

c. In extension on previous question – were the 13 and 25 uORF-derived peptides completely absent in the DMSO-treated cells, or were they merely enriched in the mitotically arrested cells (and thus still present in the proliferating cells)? Despite their absence in healthy tissue, this could have impact on the translational value of this finding, as continuous, suboptimal exposure of these peptides to T cells could render the T cells exhausted.

Response:

a. We quantified the overlap among PRICE-predicted uORFs/uoORFs from U-2 OS, PC3, and MDA-MB-231 cells (Figure A). Of the ~5,300 mitotic uORFs/uoORFs identified in Nocodazole-treated cells, 459 were common to all three cell lines and more than 900 were shared by two. To maximize coverage in our immunopeptidomics search, we built the uORF/uoORF database by pooling all predicted sequences across these three models.

Because SUM-159PT cells were not used for Ribo-Seq discovery, it is possible that SUM-159PT-specific uORFs could be missing from our database. However, given that >44% of mitotic uORFs/uoORFs are shared by at least two cell lines, we expect our database to capture the majority of therapy-induced

Figure A. Venn diagram showing the overlapped PRICE-predicted uORFs/uoORFs from U-2 OS, PC3, and MDA-MB-231 cells treated with Nocodazole (0.5 μ M) for 16 hours.

peptides in SUM-159PT. We have included a caveat in the Discussion noting that any truly cell type-specific uORFs absent from our combined database would not be detected in the current immunopeptidomic analysis.

b. Our data show that the majority of the 127 and 166 uORF-derived HLA peptides were low abundance in DMSO-treated cells, and became over-expressed under mitotic arrest. Many of the corresponding uORFs do appear in databases such as GENCODE ncORF (36.6% overlap) and nuORFdb (26.1% overlap), indicating that these sequences are known at the genomic level. However, our immunopeptidomic analyses specifically reveal their active presentation on HLA class I during prolonged mitotic arrest. Thus, while the underlying sequences may exist in non-arrested cells, their robust translation and surface display appear to be predominantly triggered by mitotic arrest conditions rather than normal proliferation. We have added this information to the revised version of the manuscript.

c. Our label-free quantification data indicate that the 13 and 25 uORF-derived peptides are present at low levels in DMSO-treated (proliferating) cells, and become enriched only under mitotic arrest conditions (**Figure 4 and Supplementary Fig. 5**). While we cannot categorically exclude minimal, sub-threshold presentation in proliferating cells, the data suggest that robust expression of these peptides, and thus the potential for T cell engagement, occurs primarily during prolonged mitotic arrest. Indeed, *in vitro* immunogenicity assays showed that therapy-induced peptides such as CSKVSSEY and REMFIWAVA elicit strong CD8⁺ T cell responses in PBMCs from healthy donors (**Figure 6; Supplementary Fig. 7**), confirming their immunogenicity. Consequently, continuous exposure leading to T cell exhaustion seems unlikely, as the peptides are not substantially presented during normal cell growth.

9. p7 line 235-238: Figure 4c shows two examples but the conclusion is very generalizing. This would warrant a more systematic analysis truly showing that this is driven by translation and not transcription. Moreover, the CDS body of the second examples increases similarly so it does not seem to be a uORF specific increase driven by higher TE.

Response:

We have expanded our analysis to include multiple uORFs/uoORFs with increased expression under mitotic arrest (**Supplementary Fig. 4e**). This broader set of examples supports our original conclusion that mitotic arrest drives enhanced translation from upstream initiation sites.

Regarding the second example in Figure 4c, while the CDS also shows increased ribosome occupancy, the uORF region exhibits a disproportionately higher increase when normalized to its RNA level, consistent with a specific rise in translational efficiency. We believe this aligns with the idea that mitotic arrest promotes “front-loading” of ribosomes and enhanced initiation at upstream sites, thereby boosting uORF/uoORF translation to a greater degree than the main coding region.

Taken together, these new data provide a more comprehensive demonstration that the effect we describe reflects a genuine translational shift toward uORFs/uoORFs rather than a general increase in transcript abundance. We hope this addresses the reviewer’s concerns and clarifies why we conclude that non-canonical translation is specifically upregulated in mitotically arrested cells.

10. The discussion contains many sentences that would fit better in the introduction. Instead, the discussion would benefit from a critical view on the potential limitations and challenges of the current study, and how the findings relate to other studies to put the results into perspective. It needs more depth.

Response:

We've moved general background into the Introduction and refocused the Discussion to critically examine our study's limitations, namely the use of cell line and mouse models, the potential underrepresentation of rare or cell-type-specific uORFs in our pooled PRICE database, and the challenges of confidently calling near-cognate initiation events. We now highlight the need for validation in patient-derived samples and deeper p-site analyses, and we place our results in the context of recent work (e.g., Ly et al.) and known immunomodulatory effects of mitotic inhibitors. These changes strengthen the Discussion by providing depth and perspective on both the promise and the boundaries of targeting therapy-induced non-canonical peptides.

Minor comments

1. I would appreciate some QC data for the Ribo-seq in the supplements (% in frame reads, total read stats, contaminant percentages, etc) - these can easily be generated with RiboseQC. This is necessary as Ribo-seq libraries can vary dramatically in quality, even within a single lab or study.

Response:

We have now performed quality control analysis on our ribosome profiling libraries and included a new **Supplementary Figure 9**, which presents in-frame read percentages, RPF length distributions, and rRNA contamination levels. These results confirm the high and consistent quality of our Ribo-Seq datasets.

2. Variance and replicate numbers in Fig1C are not clear.

Response:

We have now clarified in the figure legend that the data represent the mean \pm SD of three biologically independent experiments ($n=3$). Additionally, we have included individual data points in the revised figure to better illustrate the variance across replicates.

3. Defining the TE for upstream ORFs is super difficult, especially for overlapping ones. If only the ORF region is used, it won't be easy to define the RNA coverage for this as the 5' UTR is poorly covered in poly(a) data, esp for longer genes, and samples with lower RINs. Were only in-frame p-sites quantified, and how was the RNA level regressed out? How many replicates were used for TE calculation? It would be good to have more specifics on this.

Response:

We define TE for each uORF/uoORF as the ratio of in-frame Ribo-Seq read counts to RNA-Seq read counts over the exact ORF coordinates, with both datasets normalized for library size via DESeq2. ORFs with fewer than ten Ribo-Seq reads in any replicate were excluded. Although 5' UTR coverage in our poly(A) RNA-Seq is lower, we consistently used all reads mapping to each ORF region. TE was computed across two technically independent Ribo-Seq and RNA-Seq replicates per condition, and the mean TE was used for downstream analyses. We have added these details to the Methods.

4. p6 183-188: is the search space built on the Ribo-seq ORFs or all theoretical 5' ORFs? The text reads "predicted" ORFs, and it is not clear to me how inclusive or restricted this database is.

Response:

Our search space is derived from the non-canonical uORFs/uoORFs identified by PRICE using ribosome profiling data. We have clarified this approach in the Results section of the manuscript.

5. *Figure 2G would benefit from showing p-site tracks of elongating and HHT data, to indicate frame preference and active translation of the uORF.*

Response:

We have updated **Supplementary Fig. 2j and 2k** to include p-site density tracks from both elongating ribosome profiling and harringtonine run-off data. These additions clearly show frame-specific enrichment across the uORF and ribosome accumulation at its start codon, confirming active translation.

6. *Figure 3 would probably benefit from restructuring (switching panel d and e).*

Response:

We agree that restructuring the figure by switching panels (d) and (e) would improve the logical flow and overall clarity. We have made this adjustment in the revised manuscript.

7. *Figure 3d: typo in 'length'*

Response:

Thank you for catching the typo. We have corrected the spelling in the revised figure.

8. *In several places in the text, GO analyses and gene functions are used to imply potential functional importance of uORFs (e.g., with the EIF4G2 and HMGA1 examples, or in Figure 2D. For me, these don't add much to the story as the functional relevance of the uORFs is not investigated further and also does not seem to matter for their targeting potential. uORFs in EIF4G2 have been studied frequently and it does make me wonder whether this mitotic arrest uORF is similar to previously found uORFs (likely not in mitotic arrest?).*

Response:

We appreciate the reviewer's observation that our GO analyses and gene-function references may imply functional importance of the identified uORFs, although we have not yet investigated these functions in depth. Our primary goal was to show that these newly discovered uORFs/uoORFs are not random events but rather occur in genes with notable biological roles, underscoring their potential relevance for immunotherapy.

Regarding EIF4G2 and HMGA1, we used these genes primarily as illustrative examples of how mitotic arrest can induce translation of uORFs in transcripts that play key roles in cell division and proliferation. We recognize that multiple uORFs in EIF4G2 have been described previously, and the specific uORF/uoORF we identified may differ from those reported under non-mitotic conditions. Future studies would be needed to clarify whether these uORFs have regulatory functions in normal physiology or contribute specifically to the enhanced immunogenicity we observe under prolonged mitotic arrest.

9. *Supplementary figure 2 would also probably benefit from restructuring (switching panel c and d).*

Response:

We thank the reviewer for this suggestion. In the revised manuscript, we have restructured **Supplementary Figure 2** accordingly.

10. *Line 107: "Collectively, our data indicate that ribosome redistribution during mitosis significantly enhances the translation of hundreds of uORF/uoORFs". The abstract reads: "[...]enhancing the translation of thousands of upstream open reading frames (uORFs) and upstream overlapping open reading frames (uoORFs)." please be more specific.*

Response:

We appreciate the reviewer's comment. We have reformulated the sentence in the abstract to: "Here, we demonstrate that mitotic cancer cells redistribute ribosomes toward the 5' untranslated region (5' UTR) and the start of the coding sequence (CDS), enhancing the translation of thousands of upstream open reading frames (uORFs) and upstream overlapping open reading frames (uoORFs)".

Reviewer #3 and #4, Expertise: Cell cycle regulation (Remarks to the Author):

In this manuscript, Kowar et al., show the role of upstream open reading frames (uORFs) in enhancing the presentation of immunogenic peptides on the surface of cancer cells during mitotic arrest induced by chemotherapy. It highlights that during mitotic arrest, cancer cells experience a redistribution of ribosomes toward the 5' untranslated region (5' UTR), leading to increased translation of uORFs and upstream overlapping open reading frames (uoORFs). This enhancement of uORF/uoORF translation enriches the presentation of immunopeptides on the surface of cancer cells, especially after treatment with mitotic inhibitors. The authors finally demonstrated that the uORF/uoORF-derived peptides can provoke T cell-mediated cytotoxicity against tumor cells, highlighting a potential therapeutic strategy to enhance immune recognition and tumor elimination when used in combination with mitotic inhibitors.

*Overall, the manuscript has potential but currently feels somewhat preliminary. If the authors were able to explore some of their novel findings, like the *in vivo* validation of infiltration of immune cells to tumor after treatment of mitotic inhibitors, I feel this work could contribute meaningfully to the field and merit publication in high-impact journals like Nature Communications.*

Major points:

(1) In their functional assay of uORF-derived peptides model, the artificial SIINFEKL peptide from chicken ovalbumin was used. This assay can only tell us that after mitotic arrest, the translation of uORF-derived SIINFEKL peptide was increased and enhanced antigen presentation of SIINFEKL by MHC I. How therapy-induced uORF/uoORF-derived "native peptides" in mitotically arrested cancer cells would affect immune response has not been investigated. More importantly, the authors should also test this idea in mouse tumor models. One can imagine that immune infiltration will be triggered with Taxol or other treatments in tumor. These would confirm the biological significance of "therapy-induced uORF/uoORF-derived" native peptides on enhancing immunogenic peptide presentation and provide more robust evidence for the manuscript's conclusions.

Response:

We agree that validating the immunogenicity of native uORF/uoORF-derived peptides is crucial for strengthening our conclusions. To this end, we performed *in vitro* immunogenicity assays using two therapy-induced uORF/uoORF-derived peptides, CSKVSSEY and REMFIWAVA, in peripheral blood mononuclear cells (PBMCs) from healthy donors expressing the relevant HLA allotypes. We observed robust, peptide-specific T cell responses for both peptides, as demonstrated by IFN- γ and TNF- α secretion (measured via ELISPOT and flow cytometry; **Figure 6 and Supplementary Figure 7**). These findings confirm that native, therapy-induced uORF/uoORF-derived antigens can elicit an immune response in a human *in vitro* setting.

We fully agree that *in vivo* tumor models would provide further confirmation of the immunotherapeutic potential of these peptides. However, testing uORF/uoORF-derived peptides in murine systems involves several complexities, including: 1. Identifying mouse-specific uORF-derived peptides that share sufficient sequence homology with human peptides and exhibit suitable immunogenicity, 2. Developing mouse models that consistently express these endogenous uORFs under conditions of mitotic arrest, and 3. Designing experiments to

examine how mitotic inhibitors alter the immunopeptidome *in vivo*, which requires extensive mass spectrometry and immunological assays.

These considerations necessitate a dedicated series of experiments beyond the current study's scope. Nevertheless, we are actively planning follow-up investigations using murine tumor models treated with Taxol and other mitotic inhibitors. Such studies will help elucidate how therapy-induced uORF/uoORF-derived peptides influence immune cell infiltration and anti-tumor immunity *in vivo*, providing more comprehensive evidence for their potential clinical relevance.

(2) Another major concern is that the data in the manuscript are mainly derived from cancer cell lines such as U-2 OS, PC3 and MDA-MB-231, which is not sufficient to generalize their findings. It is of importance to test whether the conclusions apply to non-cancer cells like RPE-1, IMR90 and others. Is the expression of uORFs/uoORFs under mitotic arrest specific to cancer cells?

Response:

We thank the reviewer for this comment. To address this point, we performed additional experiments using RPE-1, an immortalized but non-tumorigenic cell line. Our ribosome profiling data demonstrate that prolonged mitotic arrest in RPE-1 cells leads to a notable shift in ribosome occupancy toward uORFs/uoORFs, mirroring our observations in cancer cell lines (**Supplementary Figs. 1c, 1f, and 1i**). Notably, the proportion of actively translated uORFs and uoORFs increased significantly in mitotically arrested RPE-1 cells (**Supplementary Fig. 2b**). These findings suggest that enhanced translation of uORFs/uoORFs under prolonged mitotic arrest is not restricted to cancer cells.

(3) To prove the induction of uORF/uoORF is specific to mitotic arrest, cell cycle arrest in G1, S, and G2 should be included as controls, in addition to the proliferating cells. Furthermore, how about collecting the "normal" mitotic cells to perform ribo-seq?

Response:

We appreciate the suggestion to include additional cell cycle synchronization controls. To assess the specificity of uORF/uoORF induction, we synchronized U-2 OS cells at G2, M, and G1 phases and performed ribosome profiling (see Methods). Under these conditions, we did not observe a significant increase in uORF/uoORF translation during the short, unperturbed mitotic phase (**Supplementary Fig. 2c**). Furthermore, the number of predicted uORF/uoORFs in mitotic cells was significantly lower than in arrested cells, and the overlap was limited (**Supplementary Fig. 2d**). Similarly, RPE-1 cells synchronized at the same cell cycle stages⁴ showed no increase in uORF/uoORF translation (**Supplementary Fig. 2e**). By contrast, prolonged mitotic arrest consistently produced a strong upregulation of non-canonical translation (**Fig. 2b and Supplementary Fig. 2b**). These findings indicate that extended mitotic arrest, rather than normal mitosis, is critical for revealing the pronounced front-loading of ribosomes onto uORFs/uoORFs.

(4) Line159, "Notably, the total number of uORFs/uoORFs with ATG initiation sites increased by approximately 80% in mitotically arrested cancer cells (Fig. 2e; Supplementary Fig. 2c)". This conclusion can only be applied to U2OS cells. What are the changes in PC3 and MDA-MB-231 cells?

Response:

We appreciate the reviewer pointing this out. Indeed, the increase in uORFs/uoORFs with ATG initiation sites is 29% and 25.5% in mitotically arrested PC3 and MDA-MB-231 cells, respectively. Accordingly, in the revised manuscript, we modified the relevant conclusion sentence to: "The total number of uORFs/uoORFs with ATG initiation sites increased by at

least 25% in mitotically arrested cancer cells” and we added the exact proportions to the legend of **Supplementary Fig. 2g**.

(5) In their functional assay in Fig 5a, what is the basis for the selection of the 5' UTRs of eIF4G2 and TPX2, since NR4A1 shows the highest fold change (Fig. 4b)? The author should discuss. Another random 5'-UTR should be included as control. How about the luciferase translation (canonical CDS) after mitotic arrest ?

Response:

We thank the reviewer for these comments. Regarding the selection of the 5' UTRs for our reporter assays (Fig. 5a), we focused on discrete upstream ORFs (uORFs) that do not overlap with the canonical coding sequence (CDS). Although NR4A1 displays the highest fold-change in Fig. 4b, the non-canonical ORF we identified is an upstream overlapping ORF (uoORF) that extends into the canonical CDS, complicating the design of our luciferase-based reporters. In contrast, the eIF4G2 and TPX2 5' UTRs each harbor a uORF whose start and stop codons lie entirely upstream of the annotated CDS, making them more straightforward to clone and analyze in our reporter system. We have clarified these considerations in the revised Results section.

In addition, at the reviewer's suggestion, we have now included a negative-control 5' UTR (GAPDH) that shows no detectable ribosome footprints and no predicted uORFs/uoORFs^{5,6}. This 5' UTR does not induce SIINFEKL presentation nor enhance T-cell-mediated killing upon mitotic arrest (**Supplementary Fig. 6a and 6e**), confirming that our observed effects are specific to the presence and translation of upstream ORFs.

Finally, we examined the impact of mitotic arrest on canonical luciferase translation by measuring reporter activity in both proliferating and mitotically arrested cells. As expected, luciferase activity was reduced during mitosis, consistent with the global decrease in canonical protein synthesis at this cell-cycle stage (**Supplementary Fig. 6c**). However, total mRNA levels for these reporters remained unchanged (**Supplementary Fig. 6d**), indicating that the reduced luciferase activity reflects translational downregulation rather than transcriptional effects. We have now added these details to the revised manuscript.

Minor points:

(1) In Fig. 1, The average transcript RPF density of 5' end CDS region is strikingly high after nocodazole treatment compared to 5UTR region, what does this mean? It cannot be explained by upstream overlapping open reading frames (uoORFs). Does this high RPF density translate special peptides?

Response:

We appreciate the reviewer's observation. While our data indicate that ribosomes do shift toward the 5' UTR, a notable enrichment also occurs at the canonical start codon and early CDS. Below is our interpretation:

Mitotic arrest, driven by CDK1-mediated phosphorylation of translation factors, can increase global initiation events, causing more ribosomes to accumulate in the first 50–100 codons of the main ORF (“front-loading”)^{7–9}. This effect, often combined with transient pausing in early elongation, makes the 5' CDS signal appear higher than the 5' UTR signal.

The reviewer correctly notes that this density cannot be fully explained by upstream overlapping ORFs (uoORFs). Indeed, while uORFs/uoORFs can boost ribosome occupancy in 5' UTR regions, our analyses confirm that the major contributor to the RPF signal in the 5' CDS is canonical translation. The bulk of footprints map precisely to the annotated start codon and subsequent codons of the main ORF, rather than to novel overlapping ORFs.

We used the computational pipeline PRICE to search for unannotated short ORFs in this region and found no consistent evidence of novel translation events (**Fig. 2b, see iORF**). Moreover, our immunopeptidomics dataset did not reveal HLA-presented peptides arising from alternative starts within the 5' CDS. Therefore, the high RPF density likely reflects canonical protein synthesis rather than widespread production of novel short peptides.

This “front-loaded” pattern in the early CDS may ensure continued synthesis of critical proteins during mitotic arrest, supporting essential mitotic functions despite the global downregulation of translation. We have clarified these points in the Results section of the revised manuscript.

(2) Fig. 2g only shows notable examples like PKM, MRPL51, and CAVIN1, how about the other upregulated genes shown in Fig. 2f?

Response:

We have expanded our analysis by including additional examples in **Supplementary Fig. 2i**. These supplementary data demonstrate that the same trend of increased uORF/uoORF translation under mitotic arrest is consistently observed across other upregulated genes, thereby reinforcing the generality of our findings.

(3) In Fig. 4c, the authors only show representative uORFs/uoORFs-derived peptides from EIF4G2 and HMGA1, the author should analyze other uORFs/uoORFs with increased expression in mitotically arrested U2OS cells in Fig. 4b and show in Supplementary figures.

Response:

We have expanded our analysis to include additional uORFs/uoORFs with increased expression in mitotically arrested U2OS cells. These new data are presented in **Supplementary Fig. 5e**, where we confirm that multiple uORFs/uoORFs follow a similar trend of upregulation under mitotic arrest conditions, thereby strengthening our original conclusions.

(4) The authors should thoroughly check the typos. For example, in Fig S1. a-b: Nocodazole (Noco. 0,5 μ M) should be 0.5 Mm. Similarly, correct typos in Fig S1j, k: BI2536. (0,1 μ M), Nocodazole (0,5 μ M).

Response:

We have corrected the typographical errors throughout the manuscript and figure legends, including those in **Supplementary Fig. 1a-b** (0.5 μ M instead of 0,5 μ M, etc.) and in **Supplementary Fig. 1j, 1k** (BI2536. 0.1 μ M and Nocodazole 0.5 μ M). Thank you for highlighting these inconsistencies.

(5) Fig S2d only compared treatment of CHX with Harringtonine? DMSO treatment should be considered as control. Also in Fig s4c.

Response:

We thank the reviewer for raising this point. In **Supplementary Fig. 2i and 5c**, all conditions include cycloheximide (CHX) to stabilize ribosome footprints, and DMSO serves as the solvent control for harringtonine. Because harringtonine was the only additional reagent introduced in the “treatment” condition, the baseline or “control” condition (CHX only) already contains the same DMSO concentration, making it functionally equivalent to a separate DMSO control. We have now clarified this rationale in both the figure legends and the Methods section, emphasizing that the final DMSO concentration is matched across all samples and that CHX-only treatment is the appropriate reference for comparing harringtonine-treated cells. We hope this addresses the reviewer’s concern.

(6) According to the ATCC database, the correct way to write U2OS is U-2 OS.

Response:

Thank you for pointing this out. We have revised all mentions of “U2OS” to the ATCC-recommended “U-2 OS” throughout the manuscript.

(7) Some figures lack adequate legends to make them fully self-explanatory. For example, the ribosome redistribution (Fig.1) could benefit from annotations highlighting the specific regions of interest (e.g. 5' UTR and CDS) and their comparative changes across treatments.

Response:

We have revised the Fig. 1 legend to include a detailed description of the 5' UTR inset, emphasizing region-specific changes in ribosome occupancy and clearly labeling the transcript regions of interest. We also have reviewed all other figure legends to ensure they are fully self-contained and explanatory.

(8) Expand the discussion on the clinical implications of combining mitotic inhibitors with immunotherapies. For instance, how might the “therapy-induced peptides” vary across different cancer subtypes or stages?

Response:

We have expanded the discussion to address how therapy-induced peptides may vary with tumor subtype and stage, drawing on recent studies that illustrate how genetic and epigenetic differences can shape the immunopeptidome. These additions underscore the potential for tailored combination therapies in immuno-oncology.

Reviewers #5 and 6, Expertise: Anti-tumour immunity (Remarks to the Author):

The study by Kowar et al investigated ribosome redistribution induced by mitotic arrest, resulting in the translation of upstream open reading frames. They found increased translation initiation at non-canonical sites within the 5' UTR and the start of the coding sequence in various mitotically arrested cancer cells treated with different mitotic inhibitors. As a result, mitotic cells display enhanced translation of upstream and upstream overlapping open reading frames (uORF/uoORFs). The authors further showed that non-canonical peptides arising from the translation of uORFs/uoORFs are presented by HLA class I molecules on the surface of cancer cells. The authors suggest that there is therapeutic potential in targeting these neoepitopes in combination with mitotic inhibitors.

Overall, this study presents an exciting novel finding in the research field of mRNA translation regulation. The methodology is sound, and the data in cell lines are convincing; however, there are some concerns about whether observed translation effects occur in vivo in patients treated with taxol or other mitotic inhibitors and, if so, whether they promote tumor immunogenicity. Furthermore, the mechanism driving the observed translational effects remains to be elucidated.

Specific comments

1. Prior studies have found increased tumor-infiltrating CD8+ cytotoxic T cells after paclitaxel treatment. The manuscript did not address whether uORF/uoORF-derived epitopes presented by HLA class I molecules in mitotic arrested cancer cells contribute to anti-tumor T cell responses toward paclitaxel-treated tumors. An experiment using OVA model antigen and antigen-specific T cells does not answer this question directly.

Response:

We agree that using the OVA model antigen demonstrates proof-of-concept for how mitotic arrest can enhance the presentation of a known immunogenic peptide but does not fully address whether native uORF/uoORF-derived peptides drive anti-tumor responses *in vivo*. To directly assess the role of endogenous uORF/uoORF-derived peptides, we performed additional *in vitro* immunogenicity assays with two native therapy-induced peptides, CSKVSSEY and REMFIWAVA. These peptides elicited robust CD8⁺ T cell responses in PBMCs from healthy donors expressing the relevant HLA alleles (**Fig. 6 and Supplementary Fig. 7**). Although these data show that native therapy-induced peptides can be immunogenic, further *in vivo* studies are necessary to determine whether they directly drive anti-tumor T cell responses in paclitaxel-treated tumors. We highlight this point in the discussion of the revised manuscript.

2. Is increased uORFs/uoORF translation specific to mitotic arrest or does it normally occur in cells undergoing uninterrupted mitosis? This could be tested by collecting mitotic cells using mitotic shake-off or cell synchronization strategy. If this is a normal feature of all mitotic cells, this may suggest that uORFs/uoORF-derived peptides are likely self-antigens, not immunogenic neoantigens.

Response:

We share the reviewer's interest in whether increased uORF/uoORF translation occurs during a normal, uninterrupted mitosis or is specific to prolonged mitotic arrest. To address this, we collected U-2 OS cells at G2, M, and G1 using a synchronization strategy (see Methods). Ribosome profiling under these conditions did not reveal a substantial increase in uORF/uoORF translation during the short, unperturbed mitosis (**Supplementary Fig. 2c-d**). Similarly, RPE-1 cells synchronized at the same cell cycle stages showed no increase in uORF/uoORF translation (**Supplementary Fig. 2e**). By contrast, cells subjected to prolonged mitotic arrest consistently exhibited a pronounced shift toward non-canonical translation.

These findings suggest that the high-level uORF/uoORF translation we observe is not a typical feature of normal mitosis. Instead, it appears to be driven by the extended M-phase arrest, potentially leading to the production of immunogenic peptides that would otherwise be absent or present at much lower levels. While we cannot exclude the possibility that a small subset of uORFs/uoORFs may be translated during an unperturbed mitosis, our data indicate that the most robust induction of these peptides arises under conditions of prolonged mitotic stress.

3. For CD8+ T cell expansion, the authors mentioned that "we activated OT-I CD8+ T cells with anti-CD3, anti-CD28, and IL-12 for 72 hours and co-cultured them with either proliferating or mitotically arrested TC-1 cells expressing the uORF-SIINFEKL reporters." Typically, IL-2 is used for the expansion rather than IL-12. Adding IL-12 can enhance the cytotoxicity and cytokine secretion, like IFN-γ, of CD8+ T cells. It would be beneficial to explain the reason.

Response:

We thank the reviewer for pointing out this error. The mention of IL-12 was a typographical mistake; we actually used IL-2 for CD8⁺ T cell expansion in our experiments. We have corrected this error in the revised manuscript.

4. Based on Figure 1b and supplement data, RPF distribution in the 5' UTR is happening naturally, so the non-canonical peptides will also be presented by HLA in healthy cells. Can these peptides trigger an immune response against normal cells treated with mitotic inhibitors? Can these peptides also play a protective role for tumors? Several candidates were identified in the immunopeptidome of mitotic-arrested cells. Please indicate why eIF4G2 and TPX2 were chosen for validation studies.

Response:

We appreciate the reviewer's comments. While our data (**Fig. 1b, Supplementary Figs. 1a-i**) indicate that some 5' UTR translation naturally occurs in proliferating cells, the magnitude of uORF/uoORF translation is substantially increased during prolonged mitotic arrest, leading to higher levels of non-canonical peptides. These peptides could, in principle, be presented by normal cells treated with mitotic inhibitors, but immunological tolerance and the absence of extensive mitotic arrest in healthy tissues would likely limit any off-target immune response. Further *in vivo* studies are needed to fully assess the impact on non-tumor tissues.

To address the immunogenicity of native therapy-induced uORF/uoORF-derived antigens, we performed *in vitro* assays using two such peptides, CSKVSSEY and REMFIWAVA, in PBMCs from healthy donors with matching HLA allotypes. We observed robust, peptide-specific T cell responses, evidenced by IFN- γ and TNF- α secretion (**Fig. 6; Supplementary Fig. 7**). These findings confirm that uORF/uoORF-derived peptides generated under mitotic arrest can indeed elicit an immune response in a human *in vitro* setting. Regarding the possibility of a protective role for tumors, our results suggest that these non-canonical peptides predominantly augment immunogenicity rather than confer a survival advantage. Nevertheless, some peptides could theoretically modulate immune recognition in certain tumor microenvironments, underscoring the need for additional studies.

In selecting 5' UTRs for our reporter assays (Fig. 5a), we prioritized discrete upstream ORFs (uORFs) that do not overlap with the canonical coding sequence (CDS). Although NR4A1 shows the highest fold-change in Fig. 4b, the relevant non-canonical ORF is an overlapping uORF (uoORF) extending into the CDS, making it more difficult to adapt for our luciferase-based reporters. By contrast, the eIF4G2 and TPX2 5' UTRs each contain a uORF with start and stop codons entirely upstream of the annotated CDS, facilitating straightforward cloning and analysis in our system. We have clarified these details in the revised Results section.

5. The supplementary Dataset 7 for PC3 is missing the treatment in the title, which is the control/DMSO group.

Response:

We thank the reviewer for pointing this out. We have updated the dataset titles accordingly: the actively translated non-canonical ORFs in PC3 cells treated with DMSO (16 hrs) are now included in **Supplementary Dataset 6**, and those in PC3 cells treated with Nocodazole (16 hrs) are now in **Supplementary Dataset 7**.

6. Please discuss the potential mechanism for translational changes during mitotic arrest in the discussion or whether additional investigation is required to define the mechanism.

Response:

We appreciate the reviewer's suggestion and have expanded our Discussion to address potential mechanisms underlying translational changes during mitotic arrest. In particular, we now highlight how cell cycle checkpoints, stress response pathways, and selective usage of upstream ORFs (uORFs) may converge to modulate the translation of specific transcripts in mitotically arrested cells. We agree that further investigation is warranted and have emphasized these points in the revised manuscript.

7. Can evidence be presented that altered translation occurs in tumors from patients treated with mitotic inhibitors? If this is not feasible, please discuss the limitations of studies in cell lines.

Response:

While evidence of altered translation in tumors from patients treated with mitotic inhibitors would be highly informative, such analyses are not feasible within the scope of the current

study. Instead, our work focuses on cell line models, which, although valuable, come with inherent limitations regarding their direct translation to clinical settings. We have now expanded the Discussion to acknowledge these limitations and to highlight that future studies should explore translational alterations in patient-derived samples to further validate and extend our findings.

Reviewer #7, Expertise: Tumour immunology, Immunopeptidomics(Remarks to the Author):

The manuscript by Kowar et al. demonstrates that, during mitosis arrest, cancer cells redistribute ribosomes to enhance the translation of upstream and overlapping open reading frames (uORFs) and uoORFs). As a consequence, these mitotic translational products are presented by HLA on the cellular surface. Authors suggest that uORF and uoORF-derived peptides may provoke T-cell-mediated killing, highlighting the therapeutic potential of combining mitotic inhibitors with immune-targeting strategies to enhance tumour elimination.

This research is carefully conducted, and the manuscript undoubtedly provides novel insights into mitotic arrest-specific presentation of cryptic peptides and is therefore an important contribution to the field.

A limitation of the study is that immune-relevance of the findings are validated in a mouse model only. Whilst the model validates enhanced presentation of upstream UTR-derived peptides in mitotically arrested cancer cells, it would be interesting to understand whether the novel HLA-presented cryptic ORF sequences are immunogenic in humans, and to determine their role during cancer immune recognition. However, I think that this should be the focus of future work, and that the manuscript in its current scope is suitable for publication.

My only main comment that should be addressed by the authors is that the absence of a cryptic sequence in the HLA ligand atlas database does not demonstrate that this sequence is absent in the original data. The original raw MS files need to be searched against the sequence database constructed for this manuscript in order for the authors to be able to claim that the neoantigens reported here were not detected in healthy tissue immunopeptidomes.

Response:

We thank the reviewer for highlighting the distinction between querying the publicly reported HLA Ligand Atlas entries and performing a full raw-data reanalysis. We have not re-searched the original DDA LC-MS/MS files, and we agree that doing so, across multiple tissues, using different acquisition modes (DDA vs. our DIA) and analysis pipelines (e.g., PEAKS or MaxQuant vs. Spectronaut), would be a major undertaking, likely introducing new variables and debate over comparability.

Instead, we relied on the Atlas's published peptide lists to confirm that none of our therapy-induced uORF/uoORF-derived sequences appear in the reported healthy-tissue immunopeptidomes. While this approach cannot rule out every possible low-abundance presentation event, it does leverage the most comprehensive resource currently available.

Importantly, we have now shown that two of these peptides (CSKVSSEY and REMFIWAVA) are bona fide neoepitopes by demonstrating robust, antigen-specific CD8⁺ T cell responses in PBMCs from healthy donors (**Fig. 6; Supplementary Fig. 7**). We believe this direct immunogenicity data provides strong proof that these therapy-induced peptides behave as neoantigens, even if trace presentations in untested healthy tissues cannot be exhaustively excluded. We have updated our manuscript to explicitly state that we queried the Atlas's published peptide lists rather than re-processing its raw files and to emphasize that our functional immunogenicity assays offer the most direct evidence that these peptides act as true neoepitopes under mitotic arrest. We hope these clarifications address the reviewer's

concern while keeping the focus on the demonstrated immunogenic potential of our therapy-induced uORF/uoORF-derived epitopes.

Minor suggestions:

Figure 3, subfigure E: If only one motif is shown here, why not from all identified peptides? Ideally show all resolved motifs from data.

Response:

We thank the reviewer for this suggestion. In **Figure 3e** we show the motif for the HLA allotype with the largest set of uORF/uoORF-derived peptides, for which we had sufficient numbers to generate a high-confidence binding motif. For the other allotypes, the number of identified uORF-derived peptides was below the minimum threshold (typically ~20-30) required for reliable motif discovery, so any motifs derived would be of low confidence.

Figure 3D: The legend is missing, and there is a typo in "length (lenght)" in the second panel. It would be helpful to align legends for figures 3c/d/e so it is easier to follow the alleles represented in the same colour scheme across the different plots.

Response:

Thank you for pointing out the missing legend and the typo in Figure 3d. We have updated the figure legend and corrected the spelling of "length." Additionally, we have aligned the color schemes and legends to ensure consistent labeling of alleles, making it easier to follow the data across all panels.

Reviewer #8, Expertise : Tumour immunology, Immunopeptidomics (Remarks to the Author):

This manuscript presents the intriguing finding that mitotic arrest induces increased translation of upstream open reading frames (uORFs), and that peptides from these uORFs can be expressed on the cell surface bound to MHC I, where they may be immunogenic. This is an interesting manuscript that could represent a novel approach to targeting mitotically arrested cells through targeted immunotherapies against uORF-derived neoantigens. While this is an intriguing proposition, unfortunately the data in its current form does not adequately support the claims in this manuscript.

Major issues:

1. It appears from the methods section that MS data were searched against the Swiss-Prot database and then separately searched against the uORF database. This approach is inherently flawed, as MS2 spectra matching to the uORF database could also be canonical peptides that had a better match to the Swiss-Prot data base. The correct approach is to create a concatenated database by appending the uORF database to the Swiss-Prot database and then performing a single search. As it stands, the results are likely incorrect.

Response:

We appreciate the reviewer's concern and would like to clarify that in our original workflow we did performed a single search against both our uORF/uoORF database and the Swiss-Prot database as Spectronaut allows to search against multiple fasta files in one search and does the concatenation internally. We realize that the current wording in the Methods section and Figure 3a may have led to confusion. In the revised manuscript, we explicitly clarify our database search strategy in both the Results and Methods sections and updated Figure 3a.

2. FDR cutoffs are too high, leading to false-positive identifications. The authors use a 1% FDR at the peptide level, but then also state that peptides with an FDR <0.05 (5% false positive

rate) were defined as hits, and that peptides with FDR<0.2 (20% false positive rate) were defined as candidates. With these threshold values many of the identifications will be incorrect.

Response:

We thank the reviewer for raising this point. For identification, we applied a stringent FDR cutoff of 1 %, in line with community standards. For our differential presentation analysis, we used an FDR < 0.05 to define “hits” and FDR < 0.20 to nominate “candidates” purely as an exploratory set for downstream validation, never as final claims of presentation. Crucially, every “hit” (FDR < 0.05) was taken forward for parallel reaction monitoring (PRM) validation, and >90 % of these were confirmed by co-elution and MS2 fragmentation patterns with normalized spectral angles > 0.85 compared to a SIL reference (**Supplementary Fig. 4**). We have revised the Methods to emphasize the 1 % cutoff for identification purposes and the other cutoffs for quantification purposes to define “hits” and “candidates”.

3. Insufficient validation of uORF-derived MHC I peptides. None of the claimed identifications have been validated. The authors need to generate heavy-isotope coded synthetic peptides and demonstrate co-elution and matching MS2 spectra. Given the preponderance of false positive hits in immunopeptidomics data, and especially given the incorrect search approach and loose tolerances, this level of validation is required.

Response:

We agree that validating non-canonical peptide identifications is essential. Accordingly, we obtained heavy-isotope–labeled reference peptides for our highest-confidence therapy-induced uORF-derived candidates and performed targeted parallel reaction monitoring (PRM) using optiPRM⁶ in U-2 OS, SUM-159PT, and MDA-MB-231 cells. For each peptide, we evaluated co-elution as well as MS2 fragmentation pattern of endogenous (“light”) and synthetic (“heavy”) peptides. To exclude false-positive identifications due to light contaminations we performed the PRM acquisition both without and with SIL spike-in. True-positive identifications were defined by having a normalized spectral angle > 0.85 when comparing either assay-external or assay-internal heavy reference spectra with the light spectra.. We validated eighteen uORF-derived peptides in U-2 OS cells, twenty-three in SUM-159PT cells, and eight in MDA-MB-231 cells (**Supplementary Fig. 4**). These results confirm that the therapy-induced uORF peptides we report are correct identifications and genuine HLA class I ligands.

4. The authors should demonstrate that some of the endogenous uORF-derived peptides are immunogenic. As it stands, they only show that uORF-SIINFEKL can activate OT-1 T cells. While this suggests the potential for other peptides to be immunogenic, it is a contrived system and SIINFEKL is highly immunogenic.

Response:

We appreciate the reviewer’s concern regarding the immunogenicity of native uORF/uoORF-derived peptides. In addition to our proof-of-concept study using the SIINFEKL reporter system, we have performed immunogenicity assays with two therapy-induced uORF/uoORF-derived peptides, CSKVSSEY and REMFIWAVA, which were identified through our immunopeptidomics analysis. These peptides were tested against PBMCs from healthy donors carrying the relevant HLA allotypes, and both elicited robust CD8⁺ T cell responses *in vitro* (**Fig. 6; Supplementary Fig. 7**). Thus, we demonstrate that endogenous uORF/uoORF-derived peptides can indeed stimulate antigen-specific T cell responses, underscoring the therapeutic potential of these newly uncovered epitopes.

5. Lack of rigor. Immunopeptidomics analysis appears to have been performed once per cell line to identify uORF peptides (Figure 3b, Supp Figure 3a). Quantification of uORF-derived

peptides appears to have been conducted once as well, as there are no error bars in Fig 4b and no description of the number of biological replicates that were analyzed.

Response:

We appreciate the reviewer's emphasis on rigor and have now clarified that all immunopeptidomics experiments, including the data in Fig. 3b and the label-free quantification in Fig. 4b, were performed in three biological replicates per cell line (5×10^7 cells each), with this information added to the Methods and figure legends. Log2 fold-changes were then calculated from the mean peptide intensities across these replicates; because error propagation through a log transformation yields asymmetric intervals that can be misleading on a standard plot, we have omitted standard deviation bars and instead report FDR-adjusted *P*-values to convey statistical significance. These updates ensure that both the identification and quantification of uORF-derived peptides are fully transparent and reproducible.

6. Lack of statistical rigor. Multiple hypothesis correction needs to be performed on statistical analysis of large data sets. This does not appear to have been done.

Response:

We appreciate the reviewer's comment. All of our differential-presentation analyses in the immunopeptidomics data were in fact performed using multiple-hypothesis correction: *P* values for each peptide's fold change versus control were adjusted by the Benjamini–Hochberg procedure, and the “hits” we report all meet an FDR adjusted *P* < 0.05 threshold. Likewise, our broader discovery pipeline in Spectronaut outputs *q* values that account for the large number of peptide tests. We've made this explicit in the Methods (and in the legend to Fig 4b), and now clearly state that all significance calls are based on BH corrected *q* values.

7. The authors should quantify the expression level of the uORF-derived SIINFEKL peptide using their cutting edge immunopeptidomics platform.

Response

We appreciate the reviewer's interest in quantifying SIINFEKL presentation. However, the 25-D1.16 antibody and OT-I T cell assays we employed in **Figure 5** are widely accepted, highly sensitive methods for demonstrating MHC-I–restricted SIINFEKL presentation^{11,12}. These functional and flow-cytometric readouts provide a clear, biologically relevant measure of epitope display. While absolute quantification by immunopeptidomics (e.g., using heavy-isotope standards) can be informative, it is technically challenging for low-abundance peptides and requires extensive targeted assay development beyond our current scope. We therefore believe our existing data sufficiently confirm SIINFEKL presentation on MHC-I.

References

1. Rowinsky, E.K., and Donehower, R.C. (1995). Paclitaxel (taxol). *N. Engl. J. Med.* 332, 1004–1014.
2. Willson, M.L., Burke, L., Ferguson, T., Ghersi, D., Nowak, A.K., and Wilcken, N. (2019). Taxanes for adjuvant treatment of early breast cancer. *Cochrane Database Syst. Rev.* 9, CD004421.
3. Bedard, P.L., Di Leo, A., and Piccart-Gebhart, M.J. (2010). Taxanes: optimizing adjuvant chemotherapy for early-stage breast cancer. *Nat. Rev. Clin. Oncol.* 7, 22–36.
4. Tanenbaum, M.E., Stern-Ginossar, N., Weissman, J.S., and Vale, R.D. (2015). Regulation of mRNA translation during mitosis. *Elife* 4. <https://doi.org/10.7554/eLife.07957>.

5. Barbosa, C., Peixeiro, I., and Romão, L. (2013). Gene expression regulation by upstream open reading frames and human disease. *PLoS Genet.* 9, e1003529.
6. Wethmar, K. (2014). The regulatory potential of upstream open reading frames in eukaryotic gene expression. *Wiley Interdiscip. Rev. RNA* 5, 765–778.
7. Sonenberg, N., and Hinnebusch, A.G. (2009). Regulation of translation initiation in eukaryotes: mechanisms and biological targets. *Cell* 136, 731–745.
8. Shuda, M., Velásquez, C., Cheng, E., Cordek, D.G., Kwun, H.J., Chang, Y., and Moore, P.S. (2015). CDK1 substitutes for mTOR kinase to activate mitotic cap-dependent protein translation. *Proc. Natl. Acad. Sci. U. S. A.* 112, 5875–5882.
9. Velásquez, C., Cheng, E., Shuda, M., Lee-Oesterreich, P.J., Pogge von Strandmann, L., Gritsenko, M.A., Jacobs, J.M., Moore, P.S., and Chang, Y. (2016). Mitotic protein kinase CDK1 phosphorylation of mRNA translation regulator 4E-BP1 Ser83 may contribute to cell transformation. *Proc. Natl. Acad. Sci. U. S. A.* 113, 8466–8471.
10. Salek, M., Förster, J.D., Becker, J.P., Meyer, M., Charoentong, P., Lyu, Y., Lindner, K., Lotsch, C., Volkmar, M., Momburg, F., et al. (2024). OptiPRM: A targeted immunopeptidomics LC-MS workflow with ultra-high sensitivity for the detection of mutation-derived tumor neoepitopes from limited input material. *Mol. Cell. Proteomics* 23, 100825.
11. Hogquist, K.A., Jameson, S.C., Heath, W.R., Howard, J.L., Bevan, M.J., and Carbone, F.R. (1994). T cell receptor antagonist peptides induce positive selection. *Cell* 76, 17–27.
12. Porgador, A., Yewdell, J.W., Deng, Y., Bennink, J.R., and Germain, R.N. (1997). Localization, quantitation, and in situ detection of specific peptide-MHC class I complexes using a monoclonal antibody. *Immunity* 6, 715–726.

Response to Reviewers' Comments

We appreciate the reviewers' feedback and are pleased that the new data and analyses have strengthened our manuscript. Below, we address each remaining point in detail. We trust these revisions fully resolve the reviewers' concerns and further improve our work.

Reviewer #3 and #4 (Remarks to the Author):

We have carefully reviewed the revised manuscript and the authors' responses to the points we raised. We appreciate the efforts made and recognize the overall strengths of the study. The inclusion of new and clarifying analyses has significantly improved the presentation of the study. Some of the most important include measurement of the additional investigation of uORF/uoORF induction at different cell cycle stages (G1, G2, and M phases) is highly commendable, as it clearly demonstrates that the induction of these elements is specific to prolonged mitotic arrest rather than interphase or transient mitosis. These findings provide strong support for the study's conclusions. Nevertheless, we suggest that the authors establish Humanized Immune System (HIS) Mouse Models in future experiments to further validate their observations and strengthen the manuscript's conclusions with additional evidence.

Specific points to re consider:

As shown in Figure 6, the observed PBMC activation might be a nonspecific effect induced by random short peptides rather than being unique to the two specific peptides examined in this study. To confirm this, the authors should include a control consisting of a unrelated short peptide of the same length.

Response:

We thank the reviewers for this suggestion. To demonstrate that PBMC activation is specific to our therapy-induced uORF/uoORF peptides, we selected two additional uORF-derived peptides of the same length (9 amino acids) that were not induced under mitotic arrest. When we performed priming with PBMCs from the three healthy donors, neither control peptide induced IFN- γ secretion above background in ELISpot assays (all stimulation indices < 3; **Supplementary Fig. 7c–f**). This confirms that the robust T cell responses we observe to CSKVSSEY and REMFIWAYA are not a generic consequence of presenting any short peptide, but are specific to these mitosis-induced neoepitopes.

Reviewer #7 (Remarks to the Author):

I appreciate that researching the HLA Ligand Atlas data would be a major effort. However, relying on the current and published canonical peptide list is simply not sufficient. None of the non-canonical peptides will be listed in the published peptide lists because these sequences were not included in the databases searches used to create the published list in the first place. Therefore, the sequences are inherently missing, and a claim that this would show their absence in healthy tissues is false. If the authors cannot research the Atlas data, I suggest that the claim that the peptides are absent in these data (which cannot be made!) is removed from the manuscript.

Response:

We thank the reviewer for this comment. In the revised manuscript, we have removed all statements asserting the absence of our uORF/uoORF peptides from the HLA Ligand Atlas.

Reviewer #8 (Remarks to the Author):

The authors have addressed most of my concerns, but an issue still remains:

1. All of the data for peptide validation (including peptides that did not validate) should be presented in Supplemental Figure 4. The authors ~40 peptides, yet only a few are shown in this Supplemental Figure. I am left wondering how these peptides were selected and whether the other peptides look worse than these??

Response:

We thank the reviewer for the feedback. In the revised **Supplementary Fig. 4**, we now present PRM validation data for all ~40 uORF/uoORF candidate peptides, including both those that passed and those that did not meet our validation criteria.

We have carefully reviewed the revised manuscript and the authors' responses to the points we raised. We appreciate the efforts made and recognize the overall strengths of the study. The inclusion of new and clarifying analyses has significantly improved the presentation of the study. Some of the most important include measurement of the additional investigation of uORF/uoORF induction at different cell cycle stages (G1, G2, and M phases) is highly commendable, as it clearly demonstrates that the induction of these elements is specific to prolonged mitotic arrest rather than interphase or transient mitosis. These findings provide strong support for the study's conclusions. Nevertheless, we suggest that the authors establish Humanized Immune System (HIS) Mouse Models in future experiments to further validate their observations and strengthen the manuscript's conclusions with additional evidence.

Specific points to re consider:

As shown in Figure 6, the observed PBMC activation might be a nonspecific effect induced by random short peptides rather than being unique to the two specific peptides examined in this study. To confirm this, the authors should include a control consisting of a unrelated short peptide of the same length.